# Optimal Transport Model Distributional Robustness

**Van-Anh Nguyen**[1]     **Trung Le**[1]     **Anh Tuan Bui**[1]     **Thanh-Toan Do**[1]
**Dinh Phung** [1,2]
[1]Department of Data Science and AI, Monash University, Australia
[2]VinAI, Vietnam
{van-anh.nguyen, trunglm, tuan.bui, toan.do, dinh.phung}@monash.edu

## Abstract

Distributional robustness is a promising framework for training deep learning models that are less vulnerable to adversarial examples and data distribution shifts. Previous works have mainly focused on exploiting distributional robustness in the data space. In this work, we explore an optimal transport-based distributional robustness framework in model spaces. Specifically, we examine a model distribution within a Wasserstein ball centered on a given model distribution that maximizes the loss. We have developed theories that enable us to learn the optimal robust center model distribution. Interestingly, our developed theories allow us to flexibly incorporate the concept of sharpness awareness into training, whether it's a single model, ensemble models, or Bayesian Neural Networks, by considering specific forms of the center model distribution. These forms include a Dirac delta distribution over a single model, a uniform distribution over several models, and a general Bayesian Neural Network. Furthermore, we demonstrate that Sharpness-Aware Minimization (SAM) is a specific case of our framework when using a Dirac delta distribution over a single model, while our framework can be seen as a probabilistic extension of SAM. To validate the effectiveness of our framework in the aforementioned settings, we conducted extensive experiments, and the results reveal remarkable improvements compared to the baselines.

## 1   Introduction

Distributional robustness (DR) is a promising framework for learning and decision-making under uncertainty, which has gained increasing attention in recent years [4, 15, 16, 6]. The primary objective of DR is to identify the worst-case data distribution within the vicinity of the ground-truth data distribution, thereby challenging the model's robustness to distributional shifts. DR has been widely applied to various fields, including semi-supervised learning [5, 11, 66], transfer learning and domain adaptation [39, 16, 69, 50, 51, 38, 52, 57], domain generalization [57, 70], and improving model robustness [57, 8, 61]. Although the principle of DR can be applied to either data space or model space, the majority of previous works on DR have primarily concentrated on exploring its applications in data space.

Sharpness-aware minimization (SAM) [18] has emerged as an effective technique for enhancing the generalization ability of deep learning models. SAM aims to find a perturbed model within the vicinity of a current model that maximizes the loss over a training set. The success of SAM and its variants [36, 33, 62] has inspired further investigation into its formulation and behavior, as evidenced by recent works such as [31, 45, 2]. While [30] empirically studied the difference in sharpness obtained by SAM [18] and SWA [26], and [46] demonstrated that SAM is an optimal Bayes relaxation of standard Bayesian inference with a normal posterior, none of the existing works have explored the connection between SAM and distributional robustness.

37th Conference on Neural Information Processing Systems (NeurIPS 2023).

In this work, we study the theoretical connection between distributional robustness in model space and sharpness-aware minimization (SAM), as they share a conceptual similarity. We examine Optimal Transport-based distributional robustness in model space by considering a Wasserstein ball centered around a model distribution and searching for the worst-case distribution that maximizes the empirical loss on a training set. By controlling the worst-case performance, it is expected to have a smaller generalization error, as demonstrated by a smaller empirical loss and sharpness. We then develop rigorous theories that suggest us the strategy to learn the center model distribution. We demonstrate the effectiveness of our framework by devising the practical methods for three cases of model distribution: **(i)** *a Dirac delta distribution over a single model*, **(ii)** *a uniform distribution over several models*, and **(iii)** *a general model distribution* (i.e., a Bayesian Neural Network [48, 56]). Furthermore, we show that SAM is a specific case of our framework when using a Dirac delta distribution over a single model, and our framework can be regarded as a probabilistic extension of SAM.

In summary, our contributions in this work are as follows:

- We propose a framework for enhancing model generalization by introducing an Optimal Transport (OT)-based model distribution robustness approach (named OT-MDR). To the best of our knowledge, this is the first work that considers distributional robustness within the model space.

- We have devised three practical methods tailored to different types of model distributions. Through extensive experiments, we demonstrate that our practical methods effectively improve the generalization of models, resulting in higher natural accuracy and better uncertainty estimation.

- Our theoretical findings reveal that our framework can be considered as a probabilistic extension of the widely-used sharpness-aware minimization (SAM) technique. In fact, SAM can be viewed as a specific case within our comprehensive framework. This observation not only explains the outperformance of our practical method over SAM but also sheds light on future research directions for improving model generalization through the perspective of distributional robustness.

## 2 Related Work

### 2.1 Distributional Robustness

Distributional Robustness (DR) is a promising framework for enhancing machine learning models in terms of robustness and generalization. The main idea behind DR is to identify the most challenging distribution that is in close proximity to a given distribution and then evaluate the model's performance on this distribution. The proximity between two distributions can be measured using either a $f$-divergence [4, 15, 16, 43, 47] or Wasserstein distance [6, 21, 35, 44, 60]. In addition, some studies [7, 61] have developed a dual form for DR, which allows it to be integrated into the training process of deep learning models. DR has been applied to a wide range of domains, including semi-supervised learning [5, 11, 66], domain adaptation [57, 50], domain generalization [57, 70], and improving model robustness [57, 8, 61].

### 2.2 Flat Minima

The generalization ability of neural networks can be improved by finding flat minimizers, which allow models to find wider local minima and increase their robustness to shifts between training and test sets [29, 55, 17]. The relationship between the width of minima and generalization ability has been studied theoretically and empirically in many works, including [24, 49, 13, 19, 53]. Various methods have been proposed to seek flat minima, such as those presented in [54, 10, 32, 27, 18]. Studies such as [32, 28, 64] have investigated the effects of different training factors, including batch-size, learning rate, gradient covariance, and dropout, on the flatness of found minima. In addition, some approaches introduce regularization terms into the loss function to pursue wide local minima, such as low-entropy penalties for softmax outputs [54] and distillation losses [68, 67, 10].

SAM [18] is a method that seeks flat regions by explicitly minimizing the worst-case loss around the current model. It has received significant attention recently for its effectiveness and scalability compared to previous methods. SAM has been successfully applied to various tasks and domains, in-

cluding meta-learning bi-level optimization [1], federated learning [59], vision models [12], language models [3], domain generalization [9], Bayesian Neural Networks [53], and multi-task learning [58]. For instance, SAM has demonstrated its capability to improve meta-learning bi-level optimization in [1], while in federated learning, SAM achieved tighter convergence rates than existing works and proposed a generalization bound for the global model [59]. Additionally, SAM has shown its ability to generalize well across different applications such as vision models, language models, domain generalization, and multi-task learning. Some recent works have attempted to enhance SAM's performance by exploiting its geometry [36, 33], minimizing surrogate gap [71], and speeding up its training time [14, 41]. Moreover, [30] empirically studied the difference in sharpness obtained by SAM and SWA [26], while [46] demonstrated that SAM is an optimal Bayes relaxation of the standard Bayesian inference with a normal posterior.

# 3 Distributional Robustness

In this section, we present the background on the OT-based distributional robustness that serves our theory development in the sequel. Distributional robustness (DR) is an emerging framework for learning and decision-making under uncertainty, which seeks the worst-case expected loss among a ball of distributions, containing all distributions that are close to the empirical distribution [20].

Here we consider a generic Polish space $S$ endowed with a distribution $\mathbb{Q}$. Let $f : S \to \mathbb{R}$ be a real-valued (risk) function and $c : S \times S \to \mathbb{R}_+$ be a cost function. Distributional robustness setting aims to find the distribution $\tilde{\mathbb{Q}}$ in the vicinity of $\mathbb{Q}$ and maximizes the risk in the $\mathbb{E}$ form [61, 7]:

$$\max_{\tilde{\mathbb{Q}}:\mathcal{W}_c(\tilde{\mathbb{Q}},\mathbb{Q})<\epsilon} \mathbb{E}_{\tilde{\mathbb{Q}}}\left[f\left(z\right)\right], \tag{1}$$

where $\epsilon > 0$ and $\mathcal{W}_c$ denotes the optimal transport (OT) or a Wasserstein distance [63] for a metric $c$, which is defined as:

$$\mathcal{W}_c\left(\tilde{\mathbb{Q}},\mathbb{Q}\right) := \inf_{\gamma\in\Gamma\left(\tilde{\mathbb{Q}},\mathbb{Q}\right)} \int cd\gamma, \tag{2}$$

where $\Gamma\left(\tilde{\mathbb{Q}},\mathbb{Q}\right)$ is the set of couplings whose marginals are $\tilde{\mathbb{Q}}$ and $\mathbb{Q}$.

With the assumption that $f \in L^1\left(\mathbb{Q}\right)$ is upper semi-continuous and the cost $c$ is a non-negative lower semi-continuous satisfying $c(z, z') = 0$ iff $z = z'$, [7] shows that the *dual* form for Eq. (1) is:

$$\min_{\lambda\geq 0}\left\{\lambda\epsilon + \mathbb{E}_{z\sim\mathbb{Q}}[\max_{z'}\left\{f\left(z'\right) - \lambda c\left(z',z\right)\right\}]\right\}. \tag{3}$$

[61] further employs a Lagrangian for Wasserstein-based uncertainty sets to arrive at a relaxed version with $\lambda \geq 0$:

$$\max_{\tilde{\mathbb{Q}}}\left\{\mathbb{E}_{\tilde{\mathbb{Q}}}\left[f\left(z\right)\right] - \lambda\mathcal{W}_c\left(\tilde{\mathbb{Q}},\mathbb{Q}\right)\right\} = \mathbb{E}_{z\sim\mathbb{Q}}[\max_{z'}\left\{f\left(z'\right) - \lambda c\left(z',z\right)\right\}]. \tag{4}$$

# 4 Proposed Framework

## 4.1 OT based Sharpness-aware Distribution Robustness

Given a family of deep nets $f_\theta$ where $\theta \in \Theta$, let $\mathbb{Q}_\phi$ with the density function $q_\phi(\theta)$ where $\phi \in \Phi$ be a family of distributions over the parameter space $\Theta$. To improve the generalization ability of the optimal $\mathbb{Q}_{\phi^*}$, we propose the following distributional robustness formulation on the model space:

$$\min_{\phi\in\Phi} \max_{\tilde{\mathbb{Q}}:\mathcal{W}_d\left(\tilde{\mathbb{Q}},\mathbb{Q}_\phi\right)\leq\rho} \mathcal{L}_S\left(\tilde{\mathbb{Q}}\right), \tag{5}$$

where $S = \{(x_1, y_1), ..., (x_N, y_N)\}$ is a training set, $d\left(\theta, \tilde{\theta}\right) = \|\theta - \tilde{\theta}\|_2^p$ $(p \geq 1)$ is a distance on the model space, and we have defined

$$\mathcal{L}_S\left(\tilde{\mathbb{Q}}\right) = \mathbb{E}_{\theta\sim\tilde{\mathbb{Q}}}\left[\mathcal{L}_S\left(\theta\right)\right] = \mathbb{E}_{\theta\sim\tilde{\mathbb{Q}}}\left[\frac{1}{N}\sum_{n=1}^N \ell\left(f_\theta\left(x_n\right), y_n\right)\right]$$

with the loss function $\ell$.

The OP in (5) seeks the most challenging model distribution $\tilde{\mathbb{Q}}$ in the WS ball around $\mathbb{Q}_\phi$ and then finds $\mathbb{Q}_\phi, \phi \in \Phi$ which minimizes the worst loss. To derive a solution for the OP in (5), we define

$$\Gamma_{\rho,\phi} = \left\{ \gamma : \gamma \in \cup_{\tilde{\mathbb{Q}}} \Gamma\left(\tilde{\mathbb{Q}}, \mathbb{Q}_\phi\right), \mathbb{E}_{(\theta,\tilde{\theta})\sim\gamma}\left[d\left(\theta, \tilde{\theta}\right)\right]^{1/p} \leq \rho \right\}.$$

Moreover, the following theorem characterizes the solution of the OP in (5).

**Theorem 4.1.** *The OP in (5) is equivalent to the following OP:*

$$\min_{\phi\in\Phi} \max_{\gamma\in\Gamma_{\rho,\phi}} \mathcal{L}_S\left(\gamma\right), \tag{6}$$

*where* $\mathcal{L}_S\left(\gamma\right) = \mathbb{E}_{(\theta,\tilde{\theta})\sim\gamma}\left[\frac{1}{N}\sum_{n=1}^{N}\ell\left(f_{\tilde{\theta}}\left(x_n\right), y_n\right)\right].$

We now need to solve the OP in (6). To make it solvable, we add the entropic regularization term as

$$\min_{\phi\in\Phi} \max_{\gamma\in\Gamma_{\rho,\phi}} \left\{ \mathcal{L}_S\left(\gamma\right) + \frac{1}{\lambda}\mathbb{H}\left(\gamma\right) \right\}, \tag{7}$$

where $\mathbb{H}\left(\gamma\right)$ returns the entropy of the distribution $\gamma$ with the trade-off parameter $1/\lambda$. We note that when $\lambda$ approaches $+\infty$, the OP in (7) becomes equivalent to the OP in (6). The following theorem indicates the solution of the OP in (7).

**Theorem 4.2.** *When $p = +\infty$, the inner max in the OP in (7) has the solution which is a distribution with the density function*

$$\gamma^*\left(\theta, \tilde{\theta}\right) = q_\phi\left(\theta\right)\gamma^*\left(\tilde{\theta}\mid\theta\right),$$

*where* $\gamma^*\left(\tilde{\theta}\mid\theta\right) = \frac{\exp\{\lambda\mathcal{L}_S(\tilde{\theta})\}}{\int_{B_\rho(\theta)}\exp\{\lambda\mathcal{L}_S(\theta')\}d\theta'}$ *,* $q_\phi\left(\theta\right)$ *is the density function of the distribution $\mathbb{Q}_\phi$, and* $B_\rho(\theta) = \{\theta' : \|\theta' - \theta\|_2 \leq \rho\}$ *is the $\rho$-ball around $\theta$.*

Referring to Theorem 4.2, the OP in (7) hence becomes:

$$\min_{\phi\in\Phi} \mathbb{E}_{\theta\sim\mathbb{Q}_\phi, \tilde{\theta}\sim\gamma^*(\tilde{\theta}|\theta)}\left[\mathcal{L}_S\left(\tilde{\theta}\right)\right]. \tag{8}$$

The OP in (8) implies that given a model distribution $\mathbb{Q}_\phi$, we sample models $\theta \sim \mathbb{Q}_\phi$. For each individual model $\theta$, we further sample the particle models $\tilde{\theta} \sim \gamma^*(\tilde{\theta}\mid\theta)$ where $\gamma^*\left(\tilde{\theta}\mid\theta\right) \propto \exp\left\{\lambda\mathcal{L}_S\left(\tilde{\theta}\right)\right\}$. Subsequently, we update $\mathbb{Q}_\phi$ to minimize the average of $\mathcal{L}_S\left(\tilde{\theta}\right)$. It is worth noting that the particle models $\tilde{\theta} \sim \gamma^*(\tilde{\theta}\mid\theta) \propto \exp\left\{\lambda\mathcal{L}_S\left(\tilde{\theta}\right)\right\}$ seek the modes of the distribution $\gamma^*(\tilde{\theta}\mid\theta)$, aiming to obtain high and highest likelihoods (i.e., $\exp\left\{\lambda\mathcal{L}_S\left(\tilde{\theta}\right)\right\}$). Additionally, in the implementation, we employ stochastic gradient Langevin dynamics (SGLD) [65] to sample the particle models $\tilde{\theta}$.

In what follows, we consider three cases where $\mathbb{Q}_\phi$ is (i) *a Dirac delta distribution over a single model*, (ii) *a uniform distribution over several models*, and (iii) *a general distribution over the model space* (i.e., a Bayesian Neural Network (BNN)) and further devise the practical methods for them.

## 4.2 Practical Methods

### 4.2.1 Single-Model OT-based Distributional Robustness

We first examine the case where $\mathbb{Q}_\phi$ is a Dirac delta distribution over a single model $\theta$, i.e., $\mathbb{Q}_\phi = \delta_\theta$ where $\delta$ is the Dirac delta distribution. Given the current model $\theta$, we sample $K$ particles $\tilde{\theta}_{1:K}$ using SGLD with only two-step sampling. To diversify the particles, in addition to adding a small Gaussian

noise to each particle, given a mini-batch $B$, for each particle $\tilde{\theta}_k$, we randomly split $B = [B_k^1, B_k^2]$ into two equal halves and update the particle models as follows:

$$\tilde{\theta}_k^1 = \theta + \rho \frac{\nabla_\theta \mathcal{L}_{B_k^1}(\theta)}{\|\nabla_\theta \mathcal{L}_{B_k^1}(\theta)\|_2} + \epsilon_k^1,$$

$$\tilde{\theta}_k^2 = \tilde{\theta}_k^1 + \rho \frac{\nabla_\theta \mathcal{L}_{B_k^2}\left(\tilde{\theta}_k^1\right)}{\|\nabla_\theta \mathcal{L}_{B_k^2}\left(\tilde{\theta}_k^1\right)\|_2} + \epsilon_k^2, \tag{9}$$

where $\mathbb{I}$ is the identity matrix and $\epsilon_k^1, \epsilon_k^2 \sim \mathcal{N}(0, \rho\mathbb{I})$.

Furthermore, we base on the particle models to update the next model as follows:

$$\theta = \theta - \frac{\eta}{K} \sum_{k=1}^{K} \nabla_\theta \mathcal{L}_B\left(\tilde{\theta}_k^2\right), \tag{10}$$

where $\eta > 0$ is a learning rate.

It is worth noting that in the update formula (9), we use different random splits $B = [B_k^1, B_k^2], k = 1, \ldots, K$ to encourage the diversity of the particles $\tilde{\theta}_{1:K}$. Moreover, we can benefit from the parallel computing to estimate these particles in parallel, which costs $|B|$ (i.e., the batch size) gradient operations. Additionally, similar to SAM [18], when computing the gradient $\nabla_\theta \mathcal{L}_{B_k^2}\left(\tilde{\theta}_k^2(\theta)\right)$, we set the corresponding Hessian matrix to the identity one.

Particularly, the gradient $\nabla_\theta \mathcal{L}_{B_k^2}\left(\tilde{\theta}_k^1\right)$ is evaluated on the second half of the current batch so that the entire batch is used for the update. Again, we can take advantage of the parallel computing to evaluate $\nabla_\theta \mathcal{L}_{B_k^2}\left(\tilde{\theta}_k^1\right), k = 1, \ldots, K$ all at once, which costs $|B|$ (i.e., the batch size) gradient operations. Eventually, with the aid of the parallel computing, the total gradient operations in our approach is $2|B|$, which is similar to SAM.

### 4.2.2 Ensemble OT-based Distributional Robustness

We now examine the case where $\mathbb{Q}_\phi$ is a uniform distribution over several models, i.e., $\mathbb{Q}_\phi = \frac{1}{M} \sum_{m=1}^{M} \delta_{\theta_m}$. In the context of ensemble learning, for each base learner $\theta_m, m = 1, \ldots, M$, we seek $K$ particle models $\tilde{\theta}_{mk}, k = 1, \ldots, K$ as in the case of single model.

$$\tilde{\theta}_{mk}^1 = \theta + \rho \frac{\nabla_\theta \mathcal{L}_{B_{mk}^1}(\theta_m)}{\|\nabla_\theta \mathcal{L}_{B_{mk}^1}(\theta_m)\|_2} + \epsilon_{mk}^1,$$

$$\tilde{\theta}_{mk}^2 = \tilde{\theta}_{mk}^1 + \rho \frac{\nabla_\theta \mathcal{L}_{B_{mk}^2}\left(\tilde{\theta}_{mk}^1\right)}{\|\nabla_\theta \mathcal{L}_{B_{mk}^2}\left(\tilde{\theta}_{mk}^1\right)\|_2} + \epsilon_{mk}^2, \tag{11}$$

where $\mathbb{I}$ is the identity matrix and $\epsilon_{mk}^1, \epsilon_{mk}^2 \sim \mathcal{N}(0, \rho\mathbb{I})$.

Furthermore, we base on the particles to update the next base learners as follows:

$$\theta_m = \theta_m - \frac{\eta}{K} \sum_{k=1}^{K} \nabla_\theta \mathcal{L}_B\left(\tilde{\theta}_{mk}^2\right). \tag{12}$$

It is worth noting that the random splits $B = [B_{mk}^1, B_{mk}^2], m = 1, \ldots, M, k = 1, \ldots, K$ of the current batch and the added Gaussian noise constitute the diversity of the base learners $\theta_{1:M}$. In our developed ensemble model, we do not invoke any term to explicitly encourage the model diversity.

### 4.2.3 BNN OT-based Distributional Robustness

We finally examine the case where $\mathbb{Q}_\phi$ is an approximate posterior or a BNN. To simplify the context, we assume that $\mathbb{Q}_\phi$ consists of Gaussian distributions $\mathcal{N}(\mu_l, \mathrm{diag}(\sigma_l^2)), l = 1, \ldots, L$ over

the weight matrices $\theta = W_{1:L}$[1]. Given $\theta = W_{1:L} \sim \mathbb{Q}_\phi$, the reparameterization trick reads $W_l = \mu_l + \text{diag}(\sigma_l)\kappa_l$ with $\kappa_l \sim \mathcal{N}(0, \mathbb{I})$. We next sample $K$ particle models $\tilde{\theta}_k = [\tilde{W}_{lk}]_{lk}, l = 1, \ldots, L$ and $k = 1, \ldots, K$ from $\gamma^*(\tilde{\theta} \mid \theta)$ as in Theorem 4.2. To sample $\tilde{\theta}_k$ in the ball around $\theta$, for each layer $l$, we indeed sample $\tilde{\mu}_{lk}$ in the ball around $\mu_l$ and then form $\tilde{W}_{lk} = \tilde{\mu}_{lk} + \text{diag}(\sigma_l)\kappa_l$. We randomly split $B = [B_k^1, B_k^2], k = 1, \ldots, K$ and update the approximate Gaussian distribution as

$$\tilde{\mu}_{lk}^1 = \mu_l + \rho \frac{\nabla_{\mu_l}\mathcal{L}_{B_k^1}([\mu_l + \text{diag}(\sigma_l)\kappa_l]_l)}{\|\nabla_{\mu_l}\mathcal{L}_{B_k^1}([\mu_l + \text{diag}(\sigma_l)\kappa_l]_l)\|_2} + \epsilon_k^1,$$

$$\tilde{\mu}_{lk}^2 = \tilde{\mu}_{lk}^1 + \rho \frac{\nabla_{\mu_l}\mathcal{L}_{B_k^2}([\tilde{\mu}_{lk}^1 + \text{diag}(\sigma_l)\kappa_l]_l)}{\|\nabla_{\mu_l}\mathcal{L}_{B_k^2}([\tilde{\mu}_{lk}^1 + \text{diag}(\sigma_l)\kappa_l]_l)\|_2} + \epsilon_k^2,$$

$$\tilde{\theta}_k = [\tilde{\mu}_{lk}^2 + \text{diag}(\sigma_l)\kappa_l]_l, k = 1, \ldots, K,$$

$$\mu = \mu - \frac{\eta}{K}\sum_{k=1}^K \nabla_\mu \mathcal{L}_B(\tilde{\theta}_k) \text{ and } \sigma = \sigma - \eta\nabla_\sigma\mathcal{L}_B(\theta),$$

where $\epsilon_k^1, \epsilon_k^2 \sim \mathcal{N}(0, \rho\mathbb{I})$.

## 4.3 Connection of SAM and OT-based Model Distribution Robustness

In what follows, we show a connection between OT-based model distributional robustness and SAM. Specifically, we prove that SAM is a specific case of OT-based distributional robustness on the model space with a particular distance metric. To depart, we first recap the formulation of the OT-based distributional robustness on the model space in (5):

$$\min_{\phi\in\Phi} \max_{\tilde{\mathbb{Q}}:\mathcal{W}_d(\tilde{\mathbb{Q}},\mathbb{Q}_\phi)\leq\rho} \mathcal{L}_S(\tilde{\mathbb{Q}}) = \min_{\phi\in\Phi} \max_{\tilde{\mathbb{Q}}:\mathcal{W}_d(\tilde{\mathbb{Q}},\mathbb{Q}_\phi)\leq\rho} \mathbb{E}_{\theta\sim\tilde{\mathbb{Q}}}[\mathcal{L}_S(\theta)]. \tag{13}$$

By linking to the dual form in (3), we reach the following equivalent OP:

$$\min_{\phi\in\Phi} \min_{\lambda>0} \left\{\lambda\rho + \mathbb{E}_{\theta\sim\mathbb{Q}_\phi}\left[\max_{\tilde{\theta}}\left\{\mathcal{L}_S(\tilde{\theta}) - \lambda d(\tilde{\theta},\theta)\right\}\right]\right\}. \tag{14}$$

Considering the simple case wherein $\mathbb{Q}_\phi = \delta_\theta$ is a Dirac delta distribution. The OPs in (14) equivalently entails

$$\min_\theta \min_{\lambda>0} \left\{\lambda\rho + \max_{\tilde{\theta}}\left\{\mathcal{L}_S(\tilde{\theta}) - \lambda d(\tilde{\theta},\theta)\right\}\right\}. \tag{15}$$

The optimization problem in (15) can be viewed as a probabilistic extension of SAM. Specifically, for each $\theta \sim \mathbb{Q}_\phi$, the inner max: $\max_{\tilde{\theta}}\left\{\mathcal{L}_S(\tilde{\theta}) - \lambda d(\tilde{\theta},\theta)\right\}$ seeks a model $\tilde{\theta}$ maximizing the loss $\mathcal{L}_S(\tilde{\theta})$ on a soft ball around $\theta$ controlled by $\lambda d(\theta,\tilde{\theta})$. Particularly, a higher value of $\lambda$ seeks the optimal $\tilde{\theta}$ in a smaller ball. Moreover, in the outer min, the term $\lambda\rho$ trades off between the value of $\lambda$ and the radius of the soft ball, aiming to find out an optimal $\lambda^*$ and the optimal model $\tilde{\theta}^*$ maximizing the loss function over an appropriate soft ball. Here we note that SAM also seeks maximizing the loss function but over the ball with the radius $\rho$ around the model $\theta$.

Interestingly, by appointing a particular distance metric between two models, we can exactly recover the SAM formulation as shown in the following theorem.

**Theorem 4.3.** *With the distance metric $d$ defined as*

$$d(\theta, \tilde{\theta}) = \begin{cases} \|\tilde{\theta} - \theta\|_2 & \|\tilde{\theta} - \theta\|_2 \leq \rho \\ +\infty & \text{otherwise} \end{cases}, \tag{16}$$

*the OPs in (13), (14) with $\mathbb{Q}_\phi = \delta_\theta$, and (15) equivalently reduce to the OP of SAM as*

$$\min_\theta \max_{\tilde{\theta}:\|\tilde{\theta}-\theta\|_2\leq\rho} \mathcal{L}_S(\tilde{\theta}).$$

---

[1]For simplicity, we absorb the biases to the weight matrices

Table 1: Classification accuracy on the CIFAR datasets of the single model setting with *one particle*. All experiments are trained three times with different random seeds.

| Dataset | Method | WideResnet28x10 | Pyramid101 | Densenet121 |
|---------|--------|-----------------|------------|-------------|
| CIFAR-10 | SAM | $96.72 \pm 0.007$ | $96.20 \pm 0.134$ | $91.16 \pm 0.240$ |
| | **OT-MDR** (Ours) | $\mathbf{96.97 \pm 0.009}$ | $\mathbf{96.61 \pm 0.063}$ | $\mathbf{91.44 \pm 0.113}$ |
| CIFAR-100 | SAM | $82.69 \pm 0.035$ | $81.26 \pm 0.636$ | $68.09 \pm 0.403$ |
| | **OT-MDR** (Ours) | $\mathbf{84.14 \pm 0.172}$ | $\mathbf{82.28 \pm 0.183}$ | $\mathbf{69.84 \pm 0.176}$ |

Table 2: Classification score on Resnet18. The results of baselines are taken from [46]

| Method | CIFAR-10 | | CIFAR-100 | |
|--------|----------|----------|-----------|----------|
| | ACC ↑ | AUROC ↑ | ACC ↑ | AUROC ↑ |
| SGD | $94.76 \pm 0.11$ | $0.926 \pm 0.006$ | $76.54 \pm 0.26$ | $0.869 \pm 0.003$ |
| SAM | $95.72 \pm 0.14$ | $0.949 \pm 0.003$ | $78.74 \pm 0.19$ | $0.887 \pm 0.003$ |
| bSAM | $96.15 \pm 0.08$ | $0.954 \pm 0.001$ | $80.22 \pm 0.28$ | $0.892 \pm 0.003$ |
| **OT-MDR** (Ours) | $\mathbf{96.59 \pm 0.07}$ | $\mathbf{0.992 \pm 0.004}$ | $\mathbf{81.23 \pm 0.13}$ | $\mathbf{0.991 \pm 0.001}$ |

Theorem 4.3 suggests that the OT-based model distributional robustness on the model space is a probabilistic relaxation of SAM in general, while SAM is also a specific crisp case of the OT-based model distributional robustness on the model space. This connection between the OT-based model distributional robustness and SAM is intriguing and may open doors to propose other sharpness-aware training approaches and leverage adversarial training [23, 42] to improve model robustness. Moreover, [46] has shown a connection between SAM and the standard Bayesian inference with a normal posterior. Using the Gaussian approximate posterior $N(\omega, \nu I)$, it is demonstrated that the maximum of the likelihood loss $L(q_\mu)$ (i.e., $\mu$ is the expectation parameter of the Gaussian $N(\omega, \nu I)$) can be lower-bounded by a relaxed-Bayes objective which is relevant to the SAM loss. Our findings are supplementary but independent of the above finding. Furthermore, by linking SAM to the OT-based model distributional robustness on the model space, we can expect to leverage the rich body theory of distributional robustness for new discoveries about improving the model generalization ability.

## 5 Experiments

In this section, we present the results of various experiments[2] to evaluate the effectiveness of our proposed method in achieving distribution robustness. These experiments are conducted in three main settings: a single model, ensemble models, and Bayesian Neural Networks. To ensure the reliability and generalizability of our findings, we employ multiple architectures and evaluate their performance using the CIFAR-10 and CIFAR-100 datasets. For each experiment, we report specific metrics that capture the performance of each model in its respective setting.

### 5.1 Experiments on a Single Model

To evaluate the performance of our proposed method for one particle training, we conducted experiments using three different architectures: WideResNet28x10, Pyramid101, and Densenet121. We compared our approach's results against models trained with SAM optimizer as our baseline. For consistency with the original SAM paper, we adopted their setting, using $\rho = 0.05$ for CIFAR-10 experiments and $\rho = 0.1$ for CIFAR-100 and report the result in Table 1. In our OT-MDR method, we chose different values of $\rho$ for each half of the mini-batch $B$, and denoted $\rho_1$ for $B_1$ and $\rho_2$ for $B_2$, where $\rho_2 = 2\rho_1$ to be simplified (this simplified setting is used in all experiments). To ensure a fair comparison with SAM, we also set $\rho_1 = 0.05$ for CIFAR-10 experiments and $\rho_1 = 0.1$ for CIFAR-100. Our approach outperformed the baseline with significant gaps, as indicated in Table 1. On average, our method achieved a 0.73% improvement on CIFAR-10 and a 1.68% improvement on CIFAR-100 compared to SAM. These results demonstrate the effectiveness of our proposed method for achieving higher accuracy in one-particle model training.

---

[2]The implementation is provided in https://github.com/anh-ntv/OT_MDR.git

Table 3: Evaluation of the ensemble **Accuracy** (%) on the CIFAR-10/100 datasets. We reproduce all baselines with the same hyperparameter for a fair comparison.

| | CIFAR-10 | | | | | CIFAR-100 | | | | |
|---|---|---|---|---|---|---|---|---|---|---|
| Method | ACC $\uparrow$ | Brier $\downarrow$ | NLL $\downarrow$ | ECE $\downarrow$ | AAC $\downarrow$ | ACC $\uparrow$ | Brier $\downarrow$ | NLL $\downarrow$ | ECE $\downarrow$ | AAC $\downarrow$ |
| | | | | | **Ensemble of five Resnet10 models** | | | | | |
| Deep Ensemble | 92.7 | 0.091 | 0.272 | 0.072 | 0.108 | 73.7 | 0.329 | 0.87 | 0.145 | 0.162 |
| Fast Geometric | 92.5 | 0.251 | 0.531 | 0.121 | 0.144 | 63.2 | 0.606 | 1.723 | 0.149 | 0.162 |
| Snapshot | 93.6 | 0.083 | 0.249 | 0.065 | 0.107 | 72.8 | 0.338 | 0.929 | 0.153 | 0.338 |
| EDST | 92.0 | 0.122 | 0.301 | 0.078 | 0.112 | 68.4 | 0.427 | 1.151 | 0.155 | 0.427 |
| DST | 93.2 | 0.102 | 0.261 | 0.067 | 0.108 | 70.8 | 0.396 | 1.076 | 0.150 | 0.396 |
| SGD | 95.1 | 0.078 | 0.264 | - | 0.108 | 75.9 | 0.346 | 1.001 | - | 0.346 |
| SAM | **95.4** | 0.073 | 0.268 | 0.050 | 0.107 | 77.7 | 0.321 | 0.892 | 0.136 | 0.321 |
| **OT-MDR** (Ours) | **95.4** | **0.069** | **0.145** | **0.021** | **0.004** | **79.1** | **0.059** | **0.745** | **0.043** | **0.054** |
| | | | | | **Ensemble of three Resnet18 models** | | | | | |
| Deep Ensemble | 93.7 | 0.079 | 0.273 | 0.064 | 0.107 | 75.4 | 0.308 | 0.822 | 0.14 | 0.155 |
| Fast Geometric | 93.3 | 0.087 | 0.261 | 0.068 | 0.108 | 72.3 | 0.344 | 0.95 | 0.15 | 0.169 |
| Snapshot | 94.8 | 0.071 | 0.27 | 0.054 | 0.108 | 75.7 | 0.311 | 0.903 | 0.147 | 0.153 |
| EDST | 92.8 | 0.113 | 0.281 | 0.074 | 0.11 | 69.6 | 0.412 | 1.123 | 0.151 | 0.197 |
| DST | 94.7 | 0.083 | 0.253 | 0.057 | 0.107 | 70.4 | 0.405 | 1.153 | 0.155 | 0.194 |
| SGD | 95.2 | 0.076 | 0.282 | - | 0.108 | 78.9 | 0.304 | 0.919 | - | 0.156 |
| SAM | 95.8 | 0.067 | 0.261 | 0.044 | 0.107 | 80.1 | 0.285 | 0.808 | 0.127 | 0.151 |
| **OT-MDR** (Ours) | **96.2** | **0.059** | **0.134** | **0.018** | **0.005** | **81.0** | **0.268** | **0.693** | **0.045** | **0.045** |
| | | | | | **Ensemble of ResNet18, MobileNet and EfficientNet** | | | | | |
| Deep Ensemble | 89.0 | 0.153 | 0.395 | 0.111 | 0.126 | 62.7 | 0.433 | 1.267 | 0.176 | 0.209 |
| DST | 93.4 | 0.102 | 0.282 | 0.070 | 0.109 | 71.7 | 0.393 | 1.066 | 0.148 | 0.187 |
| SGD | 92.6 | 0.113 | 0.317 | - | 0.112 | 72.6 | 0.403 | 1.192 | - | 0.201 |
| SAM | 93.8 | 0.094 | 0.280 | 0.060 | 0.110 | 76.4 | 0.347 | 1.005 | 0.142 | 0.177 |
| **OT-MDR** (Ours) | **94.8** | **0.078** | **0.176** | **0.021** | **0.007** | **78.3** | **0.310** | **0.828** | **0.047** | **0.063** |

We conduct experiments to compare our OT-MDR with bSAM [46], SGD, and SAM [18] on Resnet18. The results, shown in Table 2, demonstrate that the OT-MDR approach consistently outperforms all baselines by a substantial margin. Here we note that we cannot evaluate bSAM on the architectures used in Table 1 because the authors did not release the code. Instead, we run our OT-MDR with the setting mentioned in the original bSAM paper.

## 5.2   Experiments on Ensemble Models

To investigate the effectiveness of our approach in the context of a uniform distribution over a model space, we examine the ensemble inference of multiple base models trained independently. The ensemble prediction is obtained by averaging the prediction probabilities of all base models, following the standard process of ensemble methods. We compare our approach against several state-of-the-art ensemble methods, including Deep Ensembles [37], Snapshot Ensembles [25], Fast Geometric Ensemble (FGE) [22], and sparse ensembles EDST and DST [40]. In addition, we compare our approach with another ensemble method that utilizes SAM as an optimizer to improve the generalization ability, as discussed in Section 4.3. The value of $\rho$ for SAM and $\rho_1$, $\rho_2$ for OT-MDR is the same as in the single model setting.

To evaluate the performance of each method, we measure five metrics over the average prediction, which represent both predictive performance (Accuracy - ACC) and uncertainty estimation (Brier score, Negative Log-Likelihood - NLL, Expected Calibration Error - ECE, and Average across all calibrated confidence - AAC) on the CIFAR dataset, as shown in Table 3.

Notably, our OT-MDR approach consistently outperforms all baselines across all metrics, demonstrating the benefits of incorporating diversity across base models to achieve distributional robustness. Remarkably, OT-MDR even surpasses SAM, the runner-up baseline, by a significant margin, indicating a better generalization capability.

## 5.3   Experiment on Bayesian Neural Networks

We now assess the effectiveness of OT-MDR in the context of variational inference, where model parameters are sampled from a Gaussian distribution. Specifically, we apply our proposed method to

the widely-used variational technique as SGVB [34], and compare its performance with the original approach. We conduct experiments on two different architectures, Resnet10 and Resnet18 using the CIFAR dataset, and report the results in Table 4. It is clear that our approach outperforms the original SGVB method in all metrics, showcasing significant improvements. These findings underscore OT-MDR ability to increase accuracy, better calibration, and improve uncertainty estimation.

Table 4: Classification scores of approximate the Gaussian posterior on the CIFAR datasets. All experiments are trained three times with different random seeds.

| Dataset | Method | Resnet10 | | | Resnet18 | | |
|---|---|---|---|---|---|---|---|
| | | ACC ↑ | NLL ↓ | ECE ↓ | ACC ↑ | NLL ↓ | ECE ↓ |
| CIFAR-10 | SGVB | $80.52 \pm 2.10$ | $\mathbf{0.78 \pm 0.23}$ | $\mathbf{0.23 \pm 0.06}$ | $86.74 \pm 1.25$ | $0.54 \pm 0.01$ | $0.18 \pm 0.02$ |
| | **OT-MDR** (Ours) | $\mathbf{81.26 \pm 0.06}$ | $0.81 \pm 0.12$ | $0.26 \pm 0.08$ | $\mathbf{87.55 \pm 0.14}$ | $\mathbf{0.52 \pm 0.01}$ | $\mathbf{0.17 \pm 0.01}$ |
| CIFAR-100 | SGVB | $54.40 \pm 0.98$ | $1.96 \pm 0.05$ | $0.21 \pm 0.00$ | $60.91 \pm 2.31$ | $1.74 \pm 0.15$ | $0.24 \pm 0.03$ |
| | **OT-MDR** (Ours) | $\mathbf{55.33 \pm 0.11}$ | $\mathbf{1.85 \pm 0.06}$ | $\mathbf{0.18 \pm 0.03}$ | $\mathbf{63.17 \pm 0.04}$ | $\mathbf{1.55 \pm 0.05}$ | $\mathbf{0.20 \pm 0.03}$ |

## 5.4 Ablation Study

**Effect of Number of Particle Models.** For multiple-particles setting on a single model as mentioned in Section 4.2.1, we investigate the effectiveness of diversity in achieving distributional robustness. We conduct experiments using the WideResnet28x10 model on the CIFAR datasets, training with $K \in \{1, 2, 3, 4\}$ particles. The results are presented in Figure 1. Note that in this setup, we utilize the same hyper-parameters as the one-particle setting, but only train for 100 epochs to save time. Interestingly, we observe that using two-particles achieved higher accuracy compared to one particle. However, as we increase the number of particles, the difference between them also increases, resulting in worse performance. These results suggest that while diversity can be beneficial in achieving distributional robustness, increasing the number of particles beyond a certain threshold may have diminishing returns and potentially lead to performance deterioration.

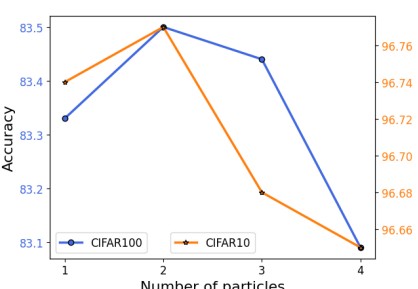

Figure 1: Multiple particle classification accuracies on the CIFAR datasets with WideResnet28x10. Note that we train 100 epochs for each experiment and report the average accuracy

**Loss landscape.** We depict the loss landscape for ensemble inference using different architectures on the CIFAR 100 dataset and make a comparison with the SAM method, which serves as our runner-up. As demonstrated in Figure 2, our approach guides the model towards a lower and flatter loss region compared to SAM, which improves the model's performance. This is important because a lower loss signifies better optimization, while a flatter region indicates improved generalization and robustness. By attaining both these characteristics, our approach enhances the model's ability to achieve high accuracy and stability (Table 3).

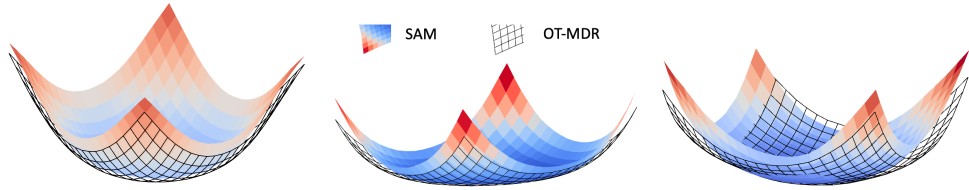

Figure 2: Comparing loss landscape of (**left**) Ensemble of 5 Resnet10, (**middle**) Ensemble of 3 Resnet18, and (**right**) Ensemble of ResNet18, MobileNet and EfficientNet on CIFAR-100 dataset training with SAM and OT-MDR. Evidently, OT-MDR leads the model to a flatter and lower loss area

# 6    Conclusion

In this paper, we explore the relationship between OT-based distributional robustness and sharpness-aware minimization (SAM), and show that SAM is a special case of our framework when a Dirac delta distribution is used over a single model. Our proposed framework can be seen as a probabilistic extension of SAM. Additionally, we extend the OT-based distributional robustness framework to propose a practical method that can be applied to (i) a Dirac delta distribution over a single model, (ii) a uniform distribution over several models, and (iii) a general distribution over the model space (i.e., a Bayesian Neural Network). To demonstrate the effectiveness of our approach, we conduct experiments that show significant improvements over the baselines. We believe that the theoretical connection between the OT-based distributional robustness and SAM could be valuable for future research, such as exploring the dual form in Eq. (15) to adapt the perturbed radius $\rho$.

**Acknowledgements.**    This work was partly supported by ARC DP23 grant DP230101176 and by the Air Force Office of Scientific Research under award number FA2386-23-1-4044.

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
