# Supplementary Material for Optimal Transport Model Distributional Robustness

Van-Anh Nguyen[1]        Trung Le[1]        Anh Tuan Bui[1]        Thanh-Toan Do[1]
Dinh Phung [1,2]
[1]Department of Data Science and AI, Monash University, Australia
[2]VinAI, Viet Nam
{van-anh.nguyen, trunglm, tuan.bui, toan.do, dinh.phung}@monash.edu

## 1 All Proofs

This section presents all proofs in our work.

$$\min_{\phi \in \Phi} \max_{\tilde{\mathbb{Q}}: \mathcal{W}_d(\tilde{\mathbb{Q}}, \mathbb{Q}_\phi) \leq \rho} \mathcal{L}_S\left(\tilde{\mathbb{Q}}\right), \tag{1}$$

where $\mathcal{W}_d\left(\tilde{\mathbb{Q}}, \mathbb{Q}_\phi\right) = \min_{\gamma \in \Gamma(\tilde{\mathbb{Q}}, \mathbb{Q}_\phi)} \mathbb{E}_{(\theta, \tilde{\theta}) \sim \gamma}\left[d\left(\theta, \tilde{\theta}\right)\right]^{1/p}$ with $d\left(\theta, \tilde{\theta}\right) = \|\tilde{\theta} - \theta\|_2^p$.

The OP in (1) seeks the most challenging model distribution $\tilde{\mathbb{Q}}$ in the WS ball around $\mathbb{Q}_\phi$ and then finds $\mathbb{Q}_\phi, \phi \in \Phi$ which minimizes the worst loss. To derive a solution for the OP in (1), we define

$$\Gamma_{\rho, \phi} = \left\{\gamma : \gamma \in \cup_{\tilde{\mathbb{Q}}} \Gamma\left(\tilde{\mathbb{Q}}, \mathbb{Q}_\phi\right), \mathbb{E}_{(\theta, \tilde{\theta}) \sim \gamma}\left[d\left(\theta, \tilde{\theta}\right)\right]^{1/p} \leq \rho\right\}.$$

**Theorem 1.1.** *The OP in (1) is equivalent to the following OP:*

$$\min_{\phi \in \Phi} \max_{\gamma \in \Gamma_{\rho, \phi}} \mathcal{L}_S\left(\gamma\right), \tag{2}$$

*where $\mathcal{L}_S\left(\gamma\right) = \mathbb{E}_{(\theta, \tilde{\theta}) \sim \gamma}\left[\frac{1}{N} \sum_{n=1}^N \ell\left(f_{\tilde{\theta}}\left(x_n\right), y_n\right)\right].$*

*Proof.* Let $\tilde{Q}^*$ be an optimal solution of the OP in (1). Let $\gamma^*$ be the optimal coupling of the WS distance $\mathcal{W}_d\left(\tilde{\mathbb{Q}}, \mathbb{Q}_\phi\right)$. Then, we have $\gamma^* \in \Gamma_{\rho, \phi}$. It follows that

$$\max_{\tilde{\mathbb{Q}}: \mathcal{W}_d(\tilde{\mathbb{Q}}, \mathbb{Q}_\phi) \leq \rho} \mathcal{L}_S\left(\tilde{\mathbb{Q}}\right) = \mathcal{L}_S\left(\tilde{\mathbb{Q}}^*\right) = \mathcal{L}_S\left(\gamma^*\right) \leq \max_{\gamma \in \Gamma_{\rho, \phi}} \mathcal{L}_S\left(\gamma\right). \tag{3}$$

Let $\gamma^* \in \Gamma_{\rho, \phi}$ be the optimal solution of (2). There exists $\tilde{Q}^*$ such that $\gamma^* \in \Gamma\left(\tilde{\mathbb{Q}}^*, \mathbb{Q}_\phi\right)$. It follows that

$$\max_{\gamma \in \Gamma_{\rho, \phi}} \mathcal{L}_S\left(\gamma\right) = \mathcal{L}_S\left(\gamma^*\right) = \mathcal{L}_S\left(\tilde{\mathbb{Q}}^*\right) \leq \max_{\tilde{\mathbb{Q}}: \mathcal{W}_d(\tilde{\mathbb{Q}}, \mathbb{Q}_\phi) \leq \rho} \mathcal{L}_S\left(\tilde{\mathbb{Q}}\right) \tag{4}$$

Leveraging Inequalities (3) and (4), we reach the conclusion.

$\square$

$$\min_{\phi \in \Phi} \max_{\gamma \in \Gamma_{\rho,\phi}} \left\{ \mathcal{L}_S(\gamma) + \frac{1}{\lambda}\mathbb{H}(\gamma) \right\}, \tag{5}$$

where $\mathbb{H}(\gamma)$ returns the entropy of the distribution $\gamma$ with the trade-off parameter $1/\lambda$. We note that when $\lambda$ approaches $+\infty$, the OP in (5) becomes equivalent to the OP in (2). The following theorem indicates the solution of the OP in (5).

**Theorem 1.2.** *When $p = +\infty$, the inner max in the OP in (5) has the solution which is a distribution with the density function*

$$\gamma^*\left(\theta, \tilde{\theta}\right) = q_\phi(\theta)\,\gamma^*\left(\tilde{\theta} \mid \theta\right),$$

*where* $\gamma^*\left(\tilde{\theta} \mid \theta\right) = \frac{\exp\{\lambda \mathcal{L}_S(\tilde{\theta})\}}{\int_{B_\rho(\theta)} \exp\{\lambda \mathcal{L}_S(\theta')\}d\theta'}$ , $q_\phi(\theta)$ *is the density function of the distribution* $\mathbb{Q}_\phi$, *and* $B_\rho(\theta) = \{\theta' : \|\theta' - \theta\|_2 \leq \rho\}$ *is the $\rho$-ball around $\theta$.*

*Proof.* Given $\gamma \in \Gamma_{\rho,\phi}$, we first prove that if $\mathbb{E}_{(\theta,\tilde{\theta})\sim\gamma}\left[d\left(\theta,\tilde{\theta}\right)\right]$ is finite $\forall p > 1$ then

$$M_\gamma := \sup_{(\theta,\tilde{\theta})\in \text{support}(\gamma)} \|\theta - \tilde{\theta}\|_2 = \lim_{p\to\infty} \mathbb{E}_{(\theta,\tilde{\theta})\sim\gamma}\left[d\left(\theta,\tilde{\theta}\right)\right]^{1/p}$$

Let denote $A_\gamma$ as the set of $(\theta,\tilde{\theta}) \in \text{support}(\gamma)$ such that $\|\theta - \tilde{\theta}\|_2 = M_\gamma$. We have

$$\mathbb{E}_{(\theta,\tilde{\theta})\sim\gamma}\left[d\left(\theta,\tilde{\theta}\right)\right]^{1/p} = \left[\int_{A_\gamma} d\left(\theta,\tilde{\theta}\right)d\gamma\left(\theta,\tilde{\theta}\right) + \int_{A_\gamma^c} d\left(\theta,\tilde{\theta}\right)d\gamma\left(\theta,\tilde{\theta}\right)\right]^{1/p}.$$

Therefore, for $(\theta,\tilde{\theta}) \in A_\gamma^c$, we have

$$\lim_{p\to\infty} \frac{d\left(\theta,\tilde{\theta}\right)}{M_\gamma^p} = \frac{\|\theta - \tilde{\theta}\|_2^p}{M_\gamma^p} = 0,$$

while for $(\theta,\tilde{\theta}) \in A_\gamma$, we have

$$\lim_{p\to\infty} \frac{d\left(\theta,\tilde{\theta}\right)}{M_\gamma^p} = \frac{\|\theta - \tilde{\theta}\|_2^p}{M_\gamma^p} = 1.$$

We derive as

$$\lim_{p\to\infty} \mathbb{E}_{(\theta,\tilde{\theta})\sim\gamma}\left[d\left(\theta,\tilde{\theta}\right)\right]^{1/p}$$

$$= M_\gamma \lim_{p\to\infty} \left[\int_{A_\gamma} \frac{d\left(\theta,\tilde{\theta}\right)}{M_\gamma^p}d\gamma\left(\theta,\tilde{\theta}\right) + \int_{A_\gamma^c} \frac{d\left(\theta,\tilde{\theta}\right)}{M_\gamma^p}d\gamma\left(\theta,\tilde{\theta}\right)\right]^{1/p}$$

$$= M_\gamma \lim_{p\to\infty} \gamma(A_\gamma)^{1/p} = M_\gamma.$$

Therefore, $\gamma \in \Gamma_{\rho,\phi}$ with $p = \infty$ is equivalent to the fact that the support set support $(\gamma)$ is the union of $B_\rho(\theta) = \left\{\theta' : \|\theta - \tilde{\theta}\|_2 \leq \rho\right\}$ with $\theta \in \text{support}(\mathbb{Q}_\phi)$.

We can equivalently turn the optimization problem of the inner max in (5) as follows:

$$\max_{\gamma \in \Gamma_{\rho,\phi}} \mathbb{E}_{(\theta,\tilde{\theta})\sim\gamma}\left[\mathcal{L}_S(\gamma)\right] + \frac{1}{\lambda}\mathbb{H}(\gamma) \tag{6}$$

$$\text{s.t. :support}(\gamma) = \cup_{\theta\in\text{support}(\mathbb{Q}_\phi)} B_\rho(\theta)$$

where $\Gamma_{\rho,\phi} = \cup_{\tilde{\mathbb{Q}}} \Gamma\left(\mathbb{Q}_\phi, \tilde{\mathbb{Q}}\right)$.

Because $\gamma \in \Gamma\left(\mathbb{Q}_\phi, \tilde{\mathbb{Q}}\right)$ for some $\tilde{\mathbb{Q}}$, we can parameterize its density function as:

$$\gamma\left(\theta, \tilde{\theta}\right) = q_\phi(\theta)\,\gamma\left(\tilde{\theta} \mid \theta\right),$$

where $q_\phi(\theta)$ is the density function of $\mathbb{Q}_\phi$ and $\gamma\left(\tilde{\theta} \mid \theta\right)$ has the support set $B_\rho(\theta)$. Please note that the constraint for $\gamma\left(\tilde{\theta} \mid \theta\right)$ is $\int_{B_\rho(\theta)} \gamma\left(\tilde{\theta} \mid \theta\right) d\tilde{\theta} = 1$.

The Lagrange function for the optimization problem in (6) is as follows:

$$\begin{aligned}
\mathcal{L} = & \int \mathcal{L}_S\left(\tilde{\theta}\right) q_\phi(\theta)\,\gamma\left(\tilde{\theta}|\theta\right) d\theta d\tilde{\theta} \\
& - \frac{1}{\lambda} \int q_\phi(\theta)\,\gamma\left(\tilde{\theta}|\theta\right) \log\left[q_\phi(\theta)\,\gamma\left(\tilde{\theta}|\theta\right)\right] d\theta d\tilde{\theta} \\
& + \int \alpha(\theta)\left[\gamma\left(\tilde{\theta} \mid \theta\right) d\tilde{\theta} - 1\right] d\tilde{\theta} d\theta,
\end{aligned}$$

where the integral w.r.t $\theta$ over on support $(\mathbb{Q}_\phi)$ and the one w.r.t. $\tilde{\theta}$ over $B_\rho(\theta)$.

Taking the derivative of $\mathcal{L}$ w.r.t. $\gamma\left(\tilde{\theta} \mid \theta\right)$ and setting it to 0, we obtain

$$0 = \mathcal{L}_S\left(\tilde{\theta}\right) q_\phi(\theta) + \alpha(\theta) - \frac{q_\phi(\theta)}{\lambda}\left[\log q_\phi(\theta) + \log \gamma\left(\tilde{\theta}|\theta\right) + 1\right].$$

$$\gamma\left(\tilde{\theta}|\theta\right) = \frac{\exp\left\{\lambda\left[\mathcal{L}_S\left(\tilde{\theta}\right) + \frac{\alpha(\theta)}{q_\phi(\theta)}\right] - 1\right\}}{q_\phi(\theta)}.$$

Taking into account $\int_{B_\rho(\theta)} \gamma\left(\tilde{\theta} \mid \theta\right) d\tilde{\theta} = 1$, we achieve

$$\int_{B_\rho(\theta)} \exp\left\{\lambda \mathcal{L}_S\left(\tilde{\theta}\right)\right\} d\tilde{\theta} = \frac{q_\phi(\theta)}{\exp\left\{\lambda \frac{\alpha(\theta)}{q_\phi(\theta)} - 1\right\}}.$$

Therefore, we arrive at

$$\gamma^*\left(\tilde{\theta}|\theta\right) = \frac{\exp\left\{\lambda \mathcal{L}_S\left(\tilde{\theta}\right)\right\}}{\int_{B_\rho(\theta)} \exp\left\{\lambda \mathcal{L}_S\left(\tilde{\theta}\right)\right\} d\tilde{\theta}}.$$

$$\gamma^*\left(\theta, \tilde{\theta}\right) = q_\phi(\theta)\,\frac{\exp\left\{\lambda \mathcal{L}_S\left(\tilde{\theta}\right)\right\}}{\int_{B_\rho(\theta)} \exp\left\{\lambda \mathcal{L}_S\left(\tilde{\theta}\right)\right\} d\tilde{\theta}}. \tag{7}$$

$\square$

$$\min_{\phi \in \Phi} \max_{\tilde{\mathbb{Q}}:\mathcal{W}_d(\tilde{\mathbb{Q}},\mathbb{Q}_\phi) \leq \rho} \mathcal{L}_S\left(\tilde{\mathbb{Q}}\right) = \min_{\phi \in \Phi} \max_{\tilde{\mathbb{Q}}:\mathcal{W}_d(\tilde{\mathbb{Q}},\mathbb{Q}_\phi) \leq \rho} \mathbb{E}_{\theta \sim \tilde{\mathbb{Q}}}\left[\mathcal{L}_S(\theta)\right]. \tag{8}$$

By linking to the dual form, we reach the following equivalent OP:

$$\min_{\phi \in \Phi} \min_{\lambda > 0}\left\{\lambda \rho + \mathbb{E}_{\theta \sim \mathbb{Q}_\phi}\left[\max_{\tilde{\theta}}\left\{\mathcal{L}_S\left(\tilde{\theta}\right) - \lambda d\left(\tilde{\theta}, \theta\right)\right\}\right]\right\}. \tag{9}$$

Considering the simple case wherein $\mathbb{Q}_\phi = \delta_\theta$ is a Dirac delta distribution. The OPs in (8) and (9) equivalently entails

$$\min_\theta \min_{\lambda > 0}\left\{\lambda \rho + \max_{\tilde{\theta}}\left\{\mathcal{L}_S\left(\tilde{\theta}\right) - \lambda d\left(\tilde{\theta}, \theta\right)\right\}\right\}. \tag{10}$$

**Theorem 1.3.** *With the distance metric $d$ defined as*

$$d\left(\theta, \tilde{\theta}\right) = \begin{cases} \|\tilde{\theta} - \theta\|_2 & \|\tilde{\theta} - \theta\|_2 \leq \rho \\ +\infty & otherwise \end{cases},$$  (11)

*the OPs in (8), (9) with $\mathbb{Q}_\phi = \delta_\theta$, and (10) equivalently reduce to the OP of SAM as*

$$\min_\theta \max_{\tilde{\theta}: \|\tilde{\theta} - \theta\|_2 \leq \rho} \mathcal{L}_S\left(\tilde{\theta}\right).$$

*Proof.* As $\mathbb{Q}_\phi = \delta_\theta$, we prove that the OP in (8) is equivalent to the OP in SAM. We start with

$$\mathcal{W}_d\left(\tilde{\mathbb{Q}}, \delta_\theta\right) = \int d\left(\tilde{\theta}, \theta\right) q\left(\tilde{\theta}\right) d\tilde{\theta}.$$

Due to the definition of the cost metric $d$, $\mathcal{W}_d\left(\tilde{\mathbb{Q}}, \delta_\theta\right) \leq \rho$ entails that $\tilde{\mathbb{Q}}$ with the density $q$ has its support set over $B_\rho(\theta)$ and $\mathcal{W}_d\left(\tilde{\mathbb{Q}}, \delta_\theta\right) = \int_{B_\rho(\theta)} \|\tilde{\theta} - \theta\|_2 q\left(\tilde{\theta}\right) d\tilde{\theta}$. Therefore, we reach

$$\max_{\tilde{\mathbb{Q}}: \mathcal{W}_d\left(\tilde{\mathbb{Q}}, \delta_\theta\right) \leq \rho} \mathcal{L}_S\left(\tilde{\mathbb{Q}}\right) = \max_{\tilde{\mathbb{Q}}: \mathcal{W}_d\left(\tilde{\mathbb{Q}}, \delta_\theta\right) \leq \rho} \int \mathcal{L}_S\left(\tilde{\theta}\right) q\left(\tilde{\theta}\right) d\tilde{\theta}$$

$$= \max_{\tilde{\mathbb{Q}}: \text{support}\left(\tilde{\mathbb{Q}}\right) = B_\rho(\theta)} \int_{B_\rho(\theta)} \mathcal{L}_S\left(\tilde{\theta}\right) q\left(\tilde{\theta}\right) d\tilde{\theta}.$$  (12)

It is obvious that the OP in (12) peaks when $\tilde{\mathbb{Q}}$ puts its all mass over the single value $\text{argmax}_{\tilde{\theta} \in B_\rho(\theta)} \mathcal{L}_S\left(\tilde{\theta}\right)$. Finally, we obtain the conclusion as

$$\max_{\tilde{\mathbb{Q}}: \mathcal{W}_d\left(\tilde{\mathbb{Q}}, \delta_\theta\right) \leq \rho} \mathcal{L}_S\left(\tilde{\mathbb{Q}}\right) = \max_{\tilde{\theta} \in B_\rho(\theta)} \mathcal{L}_S\left(\tilde{\theta}\right).$$

$\square$

## 2 Experiments

### 2.1 Experiment Setting on a Single Model

In the experiments presented in Tables 1 and 2 in the main paper, we train all models for 200 epochs using SGD with a learning rate of 0.1. We utilize a cosine schedule for adjusting the learning rate during training. To enhance the robustness of the models, we augmented the training set with basic data augmentations, including horizontal flipping, padding by four pixels, random cropping, and normalization. It's worth noting that the experiments in Table 1 utilize an input resolution of 32x32, while those in Table 2 followed the setting in [3] for the comparison with bSAM on Resnet18, which takes an input resolution of 224x224.

During our experiments, we encounter a common issue with Stochastic Gradient Langevin Dynamics (SGLD) [4] when using the noise term $\epsilon_{mk}^1, \epsilon_{mk}^2 \sim \mathcal{N}(0, \rho\mathbb{I})$ following the formulation in Formula 9. This noise can lead to a reduction in accuracy. In fact, in the original paper [4], the noise is decreased gradually across the training step from $1e-2$ to $1e-4$ or from $1e-2$ to $1e-8$. In [2], the authors also addressed this issue by reducing the noise to a very small value, from $1e-4$ to $5e-5$ for WideResNet, to ensure the convergence of SGLD. To simplify our approach while still mitigating the negative impact of the noise, we chose to fix $\epsilon_{mk}^1, \epsilon_{mk}^2 \sim \mathcal{N}(0, 0.0001)$ in most of our experiments. This choice helped to strike a balance between the diversity of models and maintaining effective convergence.

### 2.2 Experiment Setting on Ensemble Models

For experiments in Table 3 in the main paper, we follow the same data processing and model training procedures as in Table 1. We employ an ensemble approach by combining multiple models

and evaluating the scores based on the average prediction. This average prediction was obtained by aggregating the softmax predictions from all the base classifiers. Moreover, to ensure reliable uncertainty estimation, we employ calibrated uncertainty scores (Brier, NLL, ECE, and AAC). To avoid potential calibration errors that can be addressed through simple temperature scaling, as suggested in [1], we calibrate the uncertainty scores at the optimal temperature. Note that we use 15 bins for the ECE score to accurately evaluate the calibration performance.

## 2.3 Experiment Setting on Bayesian Neural Networks

In our experiments, we train Resnet10 and Resnet18 models using the Stochastic Gradient Variational Bayes (SGVB) approach. We employ the Adam optimizer with a learning rate of 0.001 and utilize a plateau schedule for training the models over 100 epochs. However, we observe that the SGVB approach yields poor performance when using different settings, which makes it challenging to scale this approach up effectively.

For our OT-MDR method, we also employ the Adam optimizer as the base optimizer to update parameters after obtaining the gradients $\nabla_\mu \mathcal{L}_B \left( \tilde{\theta}_k \right)$ and $\nabla_\sigma \mathcal{L}_B \left( \theta \right)$,. We set the hyperparameters $\rho_1 = 0.005$ and $\rho_2 = 0.01$ for all experiments with BNNs. It is important to note that in each training iteration, we sampled $\kappa_l \sim \mathcal{N}(0, \mathbb{I})$ only once, as mentioned in Section 4.2.3 of the main paper. This sampling process ensures consistency when computing the perturbation models.

# 3 Additional Ablation Studies

## 3.1 Computation complexity

The training time of OT-MDR and baselines are reported in Table 1. It is worth noting that OT-MDR takes a longer time for training since it involves calculating the gradient three times sequentially. However, for the first two times, it only needs to calculate the gradients for half of the data batch. In total, the number of gradients needed to calculate is equal to the SAM methods.

Table 1: Training time (s/epoch) for single model on CIFAR-10

| Method | WideResnet28x10 | Pyramid101 | Densenet121 |
|---|---|---|---|
| SAM | 96 | 126 | 136 |
| **OT-MDR** (Ours) | 131 | 192 | 210 |

## 3.2 Training algorithm

The training procedure[1] for OT-MDR in the single model case is outlined in Algorithm 1. To ensure diversity, we randomly split the data batch $B$ into two equal halves when computing perturbed models within each particle. This way, each particle utilizes the same data for training but in a randomized order.

Specifically, we first calculate the gradient $g_k^1$ by minimizing the loss function on the first half mini-batch $B_k^1$ to obtain the first perturbed model $\tilde{\theta}k^1$. Then, we repeat this procedure on the second mini-batch $B_k^2$ to compute the second perturbed model $\tilde{\theta}k^2$. Lastly, we calculate the third gradient $g_k^3$ by minimizing the loss function on the entire data batch $B$ and use this gradient to update the original model $\theta$.

SAM follows a similar procedure to OT-MDR, except that it skips the second perturbed model $\tilde{\theta}_k^2$ (the blue code block) and employs the entire data batch $B$ to compute the perturbed model $\theta'$ instead of just half, as in OT-MDR. In summary, both SAM and OT-MDR require an equal number of gradient calculations.

---

[1]The implementation is provided in `https://github.com/anh-ntv/OT_MDR.git`

---
**Algorithm 1** Training algorithm of OT-MDR for single model setting
---

**for** $e$ in $epochs$ **do**
    `Given data batch` $B$
    **for** $k$ in $K$ particles **do**
        $[B_k^1, B_k^2] = B$                    $\triangleright$ Randomly split the data batch $B$ into two equal halves
        $\theta' = \theta$                             $\triangleright$ Initialize model $\theta'$

        // Calculate the first perturbed model
        $g_k^1 = \nabla_{\theta'} \mathcal{L}_{B_k^1}(\theta)$
        $\epsilon_k^1 \sim \mathcal{N}(0, \rho\mathbb{I})$
        $\theta' = \theta' + \rho \frac{g_k^1}{\|g_k^1\|_2} + \epsilon_k^1$             $\triangleright$ Perturb model $\tilde{\theta}_k^1$

        **// Calculate the second perturbed model**
        $g_k^2 = \nabla_\theta \mathcal{L}_{B_k^2}(\theta')$
        $\epsilon_k^2 \sim \mathcal{N}(0, \rho\mathbb{I})$
        $\theta' = \theta' + \rho \frac{g_k^2}{\|g_k^2\|_2} + \epsilon_k^2$             $\triangleright$ Perturb model $\tilde{\theta}_k^2$

        // Calculate the actual gradient to update the model
        $g_k^3 = \nabla_\theta \mathcal{L}_B(\theta')$
    **end for**
    $\theta = \theta - \frac{\eta}{K} \sum_{k=1}^{K} g_k^3$             $\triangleright$ Update model using $g_k^3$ of K particles
**end for**

---