# OpenReview forum: "Optimal Transport Model Distributional Robustness"
_NeurIPS.cc/2023/Conference — NeurIPS 2023 poster_

### Official Review · Reviewer_X3P4 · 2023-06-16

**Soundness:** 3 good
**Presentation:** 2 fair
**Contribution:** 3 good
**Rating:** 6
**Confidence:** 3

**Summary:**

This paper explores the relationship between OT-based distributional robustness and sharpness-aware minimization (SAM), and shows that SAM is a special case of the proposed framework when a Dirac delta distribution is used over a single model. To demonstrate the effectiveness of the framework, the authors perform several experiments and show improvements over the baselines.

**Strengths:**

1. This paper is theoretically sound.
2. The experiment verifies the effectiveness of the proposed framework.

**Weaknesses:**

The practical methods stated in Section 4.2 are not easy to follow. It would be better if the authors can provide a pseudocode to help the readers understand the methods easily.

**Questions:**

1. Theorem 4.2 holds for $p=+\infty$, can the results be extended to more general settings?
2. To make the OP problem solvable, the authors resort to the entropy-regularized version of OP. However, when establishing the connection between SAM and OT-based model distribution robustness, the authors use the original version of OP. Can the authors elaborate more on this?

**Limitations:**

The practical methods stated in Section 4.2 are not easy to follow.

---

> ### Author Rebuttal · Authors · 2023-08-09
>
> We sincerely appreciate your constructive comments. We are dedicated to addressing all the questions listed below to the best of our capabilities.
>
>
> **Theorem 4.2 holds for $p=+\infty$, can the results be extended to more general settings?**
>
> Our distance of interest is $d(\theta,\tilde{\theta})=\Vert\theta-\tilde{\theta}\Vert_{2}^{p}$ and we develop a tractable solution when $p \rightarrow +\infty$ in this paper. For the case of a general $p$ or even a general distance between two models as surveyed in this paper [1], we possibly need to resort to the dual form in Eq. (3) mentioned in the main paper.
>
>
> **To make the OP problem solvable, the authors resort to the entropy-regularized version of OP. However, when establishing the connection between SAM and OT-based model distribution robustness, the authors use the original version of OP. Can the authors elaborate more on this?**
>
> Similar to other OT-based practical methods such as Sinkhorn, we need to resort to the entropy-regularized term with the trade-off $1/\lambda$ in order to develop a tractable solution for our OT-based distributional robustness on the model space. It is obvious that when $\lambda$ is sufficiently large the estimating solution can tightly approximate the ideal solution.
>
> Moreover, to establish the theoretical connection of OT-based distributional robustness on the model space and SAM, we use the original form of the proposed OT-based distributional robustness but employ a specific distance. This purpose is to demonstrate that our proposed OT-based distributional robustness is richer than SAM in terms of modelling and can be viewed as a probabilistic extension of SAM.
>
> Putting all together, from the modeling perspective, the current proposed practical solution is a relaxation of the original OT-based distributional robustness, while SAM is another specific case in the spectrum of our OT-based distributional robustness. However, we believe that this viewpoint is interesting and opens doors to develop future efficient practical methods. For instance, we can investigate the dual form in Eq. (3) using more advanced and meaningful distances between two models.
>
>
> [1] Klabunde, M., Schumacher, T., Strohmaier, M., and Lemmerich, F. (2023). Similarity of Neural Network Models: A Survey of Functional and Representational Measures. arXiv preprint arXiv:2305.06329.
>
> **The practical methods stated in Section 4.2 are not easy to follow. It would be better if the authors can provide a pseudocode to help the readers understand the methods easily.**
>
> Thanks for this advice. We will definitely provide pseudocode in the appendix of the revised version.

---

> > ### Comment · Reviewer_X3P4 · 2023-08-10
> >
> > Thank you for your response. The response addresses my concerns. I recommand accept of this paper.

---

> > > ### Author Response · Authors · 2023-08-11
> > >
> > > We are pleased to know that, and we extend our appreciation for your prompt acknowledgment.

---

### Official Review · Reviewer_WK2o · 2023-06-29

**Soundness:** 2 fair
**Presentation:** 3 good
**Contribution:** 2 fair
**Rating:** 5
**Confidence:** 4

**Summary:**

This paper proposes an optimal-transport-based distribution robustness framework on model space. The proposed theory considers the worst-case loss w.r.t a Wasserstein distance from the model distribution, which can be considered a probabilistic extension of sharpness-aware minimization (SAM).

**Strengths:**

- The paper is well-written and easy to follow.
- The proposed approach shows empirical improvements compared to SGD, SAM, DGVB, etc.

**Weaknesses:**

- The theoretical and empirical results are not surprising to me. Moreover, they do not conduct generalization analysis, which is important in learning theory.
- I do not agree with the claim that this paper is the first work that considers distribution robustness within the model space. The sharpness-related learning problem has already been well addressed in PAC Bayes theory. The counterpart of the worst-case awareness studied in this paper can be found in [1]. And other Wasserstein or more general theories can be referred to [2] and [3].
- The single-model, ensemble, and BNN cases are also common algorithm realizations in the PAC Bayes community.

>[1] Jiang, Yiding, Behnam Neyshabur, Hossein Mobahi, Dilip Krishnan, and Samy Bengio. "Fantastic Generalization Measures and Where to Find Them." In *International Conference on Learning Representations*.

>[2] Amit, R., Epstein, B., Moran, S. and Meir, R., Integral Probability Metrics PAC-Bayes Bounds. In *Advances in Neural Information Processing Systems*.

>[3] Haddouche, Maxime, and Benjamin Guedj. "Wasserstein PAC-Bayes Learning: Exploiting Optimisation Guarantees to Explain Generalisation." (2023).

**Questions:**

I suggest the author fully discuss the aforementioned related works and improve their methods or add comparisons to corresponding algorithms PAC Bayes community.

**Limitations:**

No, I suggest the authors add a section to discuss the limitation of the proposed algorithm, e.g., the increment of computation complexity of BNN and the multiple particle setting.

---

> ### Author Rebuttal · Authors · 2023-08-10
>
> We sincerely appreciate your constructive comments. We are dedicated to addressing all the questions listed below to the best of our capabilities.
>
> **Do not conduct generalization analysis, which is important in learning theory**
>
> Thanks for this comment. Generalization bound is an interesting theoretical development that strengthens our work. We will definitely consider it in future development and the papers [1,2] suggested by you possibly shed light for us to develop the bounds.
>
> In the scope of this research, we aim to find out theoretical connections between OT-based distributional robustness on a model space and sharpness-aware minimization, which are further harvested to propose practical methods on single models, ensemble models, and Bayesian neural networks.
>
> **Do not agree with the claim that this paper is the first work that considers distribution robustness within the model space.**
>
>
> The work [1] trained more than 10,000 models over two image classification datasets, namely, CIFAR-10 and Street View House Numbers (SVHN). In
> order to create diverge generalization behaviors, they carefully varied hyper-parameters that influence generalization and selected multiple optimization algorithms with different stopping criteria for training convergence. They finally investigated 40 complexity
> measures to reach conclusions. Although the empirical conclusions from this paper are interesting and beneficial, we cannot see any distributional robustness applied to model space.
>
> The work [2] is a solid study of PAC-Bayes generalization bound for the gap between general and empirical losses. Specifically, this work developed the total variation metric and the Wasserstein (WS) distance bounds. The work [3] further developed WS-style PAC-Bayes generalization bound for the data-free and data-dependent priors.
>
> To summarize, we agree that these works are related to ours, hence we will definitely mention and discuss them in the related work. However, we respectfully disagree with the reviewer the point: _these works studied WS-based distributional robustness on the model space, hence affecting the novelty of our work_.
>
> [1] Jiang, Y. et al.,  "Fantastic Generalization Measures and Where to Find Them." In International Conference on Learning Representations.
>
> [2] Amit, R. et al., Integral Probability Metrics PAC-Bayes Bounds. In Advances in Neural Information Processing Systems.
>
> [3] Haddouche, M. et al.. "Wasserstein PAC-Bayes Learning: Exploiting Optimisation Guarantees to Explain Generalisation." (2023).
>
>
> **Computation complexity**
>
> We would like to report the training time of our method and baselines (Table 4 of the attached pdf). It is with noting that our method takes longer time for training since it involves calculating the gradient three times sequentially. However, for the first two times, we only need to calculate the gradients for half of the data batch. In total, the number of gradients needs to calculate is equal to the SAM method.

---

> > ### Author Response · Authors · 2023-08-17
> >
> > Dear Reviewer,
> >
> > Could you please look at our rebuttal and kindly let us know your thought? We really appreciate your time and effort to review our paper.
> >
> > Regards,

---

> > ### Comment · Reviewer_WK2o · 2023-08-17
> >
> > Thanks for the authors' responses. I have been a little busy recently. So sorry for the late reply.
> >
> > * I am afraid that I cannot agree with your discussion on [1]. I mentioned [1] since it interprets that PAC-Bayes can capture sharpness in the expected sense compared to conventional worst-case sharpness in Sec. 4.3 (Success of Sharpness-Based Measures). Moreover, it uses the PAC-Bayesian framework to construct generalization bounds for worst-case sharpness bound (Eq.(50)) presented in Sec.C.3 (Flatness-based Measures).
> >
> > * Use PAC-Bayes bound to derive a similar optimization objective in your paper.
> >     - A typic Wasserstein-based PAC-Bayes Bound has the following form:
> >     $\mathbb{E}\_{\theta \sim \mathbb{Q}\_{\phi}} \mathcal{L}\_{\mu}(\theta) \leq \mathbb{E}\_{\theta \sim \mathbb{Q}\_{\phi} }\mathcal{L}\_{S}(\theta) + \sqrt{c_1(m, \delta)\cdot W_1(\mathbb{Q}\_{\phi}, \tilde{\mathbb{Q}}) + c_2(m,\delta)}$,
> > where $c_1, c_2$ are some sample complexity functions, $\tilde{\mathbb{Q}}$ is prior distribution of on the model space and $\mathbb{Q}_{\phi}$ is the posterior.
> >     - Based on the bound, we often set the optimization objective as
> > $\min_{\phi} \{\mathbb{E}\_{\theta \sim \mathbb{Q}\_{\phi} }\mathcal{L}\_{S}(\theta) + \lambda W_1(\mathbb{Q}\_{\phi}, \tilde{\mathbb{Q}})\}$.
> >
> >     - In PAC-Bayes community, $\tilde{\mathbb{Q}}$ is often some known prior.  But if we try to find the optimal prior,  we can have $\min_{\phi} \min_{\tilde{\mathbb{Q}}} \\{\mathbb{E}\_{\theta \sim \mathbb{Q}\_{\phi} }\mathcal{L}_{S}(\theta) + \lambda W_1(\mathbb{Q}\_{\phi}, \tilde{\mathbb{Q}})\\}$.
> >
> >     - If we set a constraint for $W_1(\mathbb{Q}_{\phi}, \tilde{\mathbb{Q}}) \leq \rho$, we can have the following objective: $\min\_{\phi}\min\_{\lambda\geq 0}\\{\lambda \rho + \max\_{\tilde{\mathbb{Q}}} \\{\mathbb{E}\_{\theta \sim \mathbb{Q}\_{\phi} }\mathcal{L}\_{S}(\theta) - \lambda W_1(\mathbb{Q}\_{\phi}, \tilde{\mathbb{Q}})\\}\\}\ $, which is the same as your eq(14).
> > These are what I expected from your discussion.
> > And that's why I said the results are not surprising to me and I do not think the paper is the first work that considers distribution robustness within the model space.
> >
> > However, since DRO and PAC-Bayes are considered different domains, and the paper has other experimental contributions, I will discuss them with AC and other reviewers to decide whether I will modify the score.

---

> > > ### Author Response · Authors · 2023-08-18
> > > **Response to the reviewer's further derivations**
> > >
> > > We thank the reviewer for providing detailed comments. While we value your input, we respectfully present our counterarguments and responses to your points and derivations as outlined below.
> > >
> > > - We agree with the reviewer up to $\\text{min}\_{\\phi}\\mathbb{E}\_{\\theta\\sim\\mathbb{Q}\_{\\phi}}\\mathcal{L}\_{S}\\left(\\theta\\right)+\\lambda W\_{1}\\left(\\mathbb{Q}\_{\\phi},\\tilde{\\mathbb{Q}}\\right)$.
> > >
> > > - We consider $\\text{min}\_{\\phi}\\mathbb{E}\_{\\theta\\sim\\mathbb{Q}\_{\\phi}}\\mathcal{L}\_{S}\\left(\\theta\\right)$ **(Eq. 1)**.
> > >
> > > - However, the next one $\\text{min}\_{\\phi}\\min\_{\\tilde{\\mathbb{Q}}}\\left\\{ \\mathbb{E}\_{\\theta\\sim\\mathbb{Q}\_{\\phi}}\\mathcal{L}\_{S}\\left(\\theta\\right)+\\lambda W\_{1}\\left(\\mathbb{Q}\_{\\phi},\\tilde{\\mathbb{Q}}\\right)\\right\\}$ **(Eq. 2)** does not make any further progress and does not have the flavor of distributional robustness. The reason is that Eq. (1) and Eq. (2) are equivalent because one can easily verify that if $\\mathbb{Q}^*$ is an optimal solution of Eq. (1) then $\\left(\\mathbb{Q}^*, \\mathbb{Q}^* \\right)$ is an optimal solution of Eq. (2).
> > >
> > > - We now look into $\\min\_{\\phi}\\min\_{\\lambda\\geq0}\\left\\{ \\lambda\\rho+\\max\_{\\tilde{\\mathbb{Q}}}\\left\\{ \\mathbb{E}\_{\\theta\\sim\\mathbb{Q}\_{\\phi}}\\mathcal{L}\_{S}\\left(\\theta\\right)-\\lambda W\_{1}\\left(\\mathbb{Q}\_{\\phi},\\tilde{\\mathbb{Q}}\\right)\\right\\} \\right\\} $ **(Eq. (3))**. It is very obvious that $\\max\_{\\tilde{\\mathbb{Q}}}\\left\\{ \\mathbb{E}\_{\\theta\\sim\\mathbb{Q}\_{\\phi}}\\mathcal{L}\_{S}\\left(\\theta\\right)-\\lambda W\_{1}\\left(\\mathbb{Q}\_{\\phi},\\tilde{\\mathbb{Q}}\\right)\\right\\} =\\mathbb{E}\_{\\theta\\sim\\mathbb{Q}\_{\\phi}}\\mathcal{L}\_{S}\\left(\\theta\\right) $, obtaining when $\\tilde{\\mathbb{Q}}=\\mathbb{Q}\_{\\phi}$, hence the OP in (3) can be simplified to  $\\min\_{\\phi}\\min\_{\\lambda\\geq0}\\left\\{ \\lambda\\rho+\\mathbb{E}\_{\\theta\\sim\\mathbb{Q}\_{\\phi}}\\mathcal{L}\_{S}\\left(\\theta\\right)\\right\\} $. This can be further minimized to $\\min\_{\\phi}\\mathbb{E}\_{\\theta\\sim\\mathbb{Q}\_{\\phi}}\\mathcal{L}\_{S}\\left(\\theta\\right)$ **Eq. (1)**, obtaining with $\\lambda = 0$.
> > >
> > > - Therefore, Eq. (3) does not have any flavor of distributional robustness. Certainly, it cannot be equivalent to our Eq. (14) which has a clear flavor of distributional robustness. Furthermore, in our derivations for both main theories and practical methodologies, we depart from Eq. (5),  which is a stronger and more comprehensive formulation than Equation (14).
> > >
> > >
> > > To summarize, we cannot see any flavor of distributional robustness within your derivations in derivations Eqs (1, 2, 3), although they look interesting. Additionally, the papers [1], [2], and [3] you referenced do not explicitly address or mention distributional robustness in the model space. We believe that it should not require the readers to infer the things that **were not mentioned explicitly in the papers**.

---

> > > > ### Comment · Reviewer_WK2o · 2023-08-18
> > > >
> > > > Sorry for the typo, it should be $\min\_{\phi}\min\_{\lambda\geq 0}\\{\lambda \rho + \max\_{\tilde{\mathbb{Q}}} \\{\mathbb{E}\_{\theta \sim \tilde{\mathbb{Q}} }\mathcal{L}\_{S}(\theta) - \lambda W_1(\mathbb{Q}\_{\phi}, \tilde{\mathbb{Q}})\\}\\}\$.
> > > >
> > > > When $W_1(\mathbb{Q}\_{\phi}, \tilde{\mathbb{Q}}) \leq \rho$, we have $\lambda =0$, and the objective becomes $\min\_{\phi}\max\_{\tilde{\mathbb{Q}}:W_1(\tilde{\mathbb{Q}}, \mathbb{Q}) \leq \rho} \mathbb{E}\_{\theta \sim \tilde{\mathbb{Q}} }\mathcal{L}\_{S}(\theta) $ as your eq.(13).
> > > >
> > > > When $W_1(\mathbb{Q}\_{\phi}, \tilde{\mathbb{Q}}) > \rho$, we have $\lambda \rightarrow \infty$, the objective becomes $\min_{\tilde{\mathbb{Q}}} W_1(\tilde{\mathbb{Q}}, \mathbb{Q}\_\phi)$.
> > > >
> > > > [1,2,3] do not mention distribution robustness, however, the sharpness-awareness has been mentioned and the related flat minima has been discussed in your paper.
> > > >
> > > > I do agree that maybe different domain has different terminology. So I will discuss these with AC and other reviewers.

---

> > > > > ### Author Response · Authors · 2023-08-19
> > > > >
> > > > > Thanks for your acute discussion which certainly benefits our future research. However, we still maintain our viewpoint that [1, 2, 3] **did not discuss** distribution robustness on a model space.
> > > > >
> > > > > [1] is an empirical paper to study what factors influence the generalization ability. The Wasserstein-based PAC-Bayes bounds developed in [2,3] have the common form of
> > > > >
> > > > > $\\mathbb{E}\_{\\theta\\sim\\mathbb{Q}\_{\\phi}}\\mathcal{L}\_{\\mu}\\left(\\theta\\right)\\leq\\mathbb{E}\_{\\theta\\sim\\mathbb{Q}\_{\\phi}}\\mathcal{L}\_{S}\\left(\\theta\\right)+\\sqrt{c\_{1}(m,\\delta)W\_{1}\\left(\\mathbb{Q}\_{\\phi},\\tilde{\\mathbb{Q}}\\right)+c\_{2}(m,\\delta)},$
> > > > >
> > > > > which suggests us the following OP
> > > > >
> > > > > $\\text{min}\_{\\phi}\\mathbb{E}\_{\\theta\\sim\\mathbb{Q}\_{\\phi}}\\mathcal{L}\_{S}\\left(\\theta\\right)+\\lambda W\_{1}\\left(\\mathbb{Q}\_{\\phi},\\tilde{\\mathbb{Q}}\\right)$
> > > > >
> > > > > Basically, similar to other KL-based PAC-Bayes bounds, this states that we aim to learn the model distribution $\\mathbb{Q}\_\\phi$ that minimizes the empirical loss and stays as closest as possible to the prior $\\tilde{\\mathbb{Q}}$. Moreover, if one considers the prior $\\tilde{\\mathbb{Q}}$ as a variable to minimize: $\\text{min}\_{\\phi}\\min\_{\\tilde{\\mathbb{Q}}}\\left\\{ \\mathbb{E}\_{\\theta\\sim\\mathbb{Q}\_{\\phi}}\\mathcal{L}\_{S}\\left(\\theta\\right)+\\lambda W\_{1}\\left(\\mathbb{Q}\_{\\phi},\\tilde{\\mathbb{Q}}\\right)\\right\\} $, this is only equivalent to $\\text{min}\_{\\phi}\\mathbb{E}\_{\\theta\\sim\\mathbb{Q}\_{\\phi}}\\mathcal{L}\_{S}\\left(\\theta\\right)$.
> > > > >
> > > > > Your subsequently suggested OP
> > > > >
> > > > > $\\min\_{\\phi}\\min\_{\\lambda\\geq0}\\left\\{ \\lambda\\rho+\\max\_{\\tilde{\\mathbb{Q}}}\\left\\{ \\mathbb{E}\_{\\theta\\sim\\tilde{\\mathbb{Q}}}\\mathcal{L}\_{S}\\left(\\theta\\right)-\\lambda W\_{1}\\left(\\mathbb{Q}\_{\\phi},\\tilde{\\mathbb{Q}}\\right)\\right\\} \\right\\} $
> > > > >
> > > > > can be reduced to
> > > > > $\\min\_{\\phi}\\max\_{\\tilde{\\mathbb{Q}}}\\mathbb{E}\_{\\theta\\sim\\tilde{\\mathbb{Q}}}\\mathcal{L}\_{S}\\left(\\theta\\right)$ if we search $\\tilde{\\mathbb{Q}}: W\_{1}\\left(\\tilde{\\mathbb{Q}},\\mathbb{Q}\_{\\phi}\\right) \\leq \\rho$.
> > > > >
> > > > > It is obvious that the subsequently suggested OP is not equivalent to the suggested OP from PAC-Bayes bounds and we cannot see the pathway to connect them. More importantly, our main contribution in this paper is to theoretically turn the distributional robustness over a model space into practical methods that can be widely applied to several situations. Furthermore, we also point out the connection between distributional robustness and sharpness-aware minimization which we believe would benefit future research in developing new methods for improving the model generalization ability.
> > > > >
> > > > > Last but not least, your thoughtful discussion hints us a further question for our future research: _investigation of the connection between the WS-based PAC-Bayes bounds and the WS-based distributional robustness_. We really appreciate it.
> > > > >
> > > > > Finally, thanks for introducing us to the excellent papers [1, 2, 3]. We will definitely discuss them in the revised version.

---

> > > > > > ### Comment · Reviewer_WK2o · 2023-08-20
> > > > > > **Response**
> > > > > >
> > > > > > Thanks for the authors' reply.
> > > > > >
> > > > > > * I want to clarify some unclear or incorrect points in my previous responses. In fact, the prior $\tilde{\mathbb{Q}}$ should be data-free in PAC-Bayes, and the theoretical optimal should be $\tilde{\mathbb{Q}} = \mathbb{E}\_S \mathbb{Q}\_{\phi}$.
> > > > > > Then we need $\min_{\phi} \\{\mathbb{E}\_{\theta \sim \mathbb{Q}\_{\phi} }\mathcal{L}_{S}(\theta) + \lambda W_1(\mathbb{Q}\_{\phi}, \tilde{\mathbb{Q}})\\}$. The robustness reflects on controlling the distance of the posterior from the prior (stability) and simultaneously minimizing the empirical risks.
> > > > > >
> > > > > > * However, you are correct that this objective cannot be straightforwardly connected to the second objective, which reflects the DRO. And considering these are in two different research domains, I raised my score accordingly.

---

> > > > > > > ### Author Response · Authors · 2023-08-20
> > > > > > >
> > > > > > > Thank you for acknowledging our contribution and valuable discussion. We will definitely add more detail in the revised version.

---

### Official Review · Reviewer_NFva · 2023-07-06

**Soundness:** 2 fair
**Presentation:** 3 good
**Contribution:** 3 good
**Rating:** 6
**Confidence:** 3

**Summary:**

This paper explores the optimal transport distributional robustness framework in the model parameters spaces, and provides practical algorithms to solve the corresponding optimization problems. This flexible framework can be used to train a single model, ensemble models and Bayesian neural networks in a sharpness-aware fashion. The authors additionally establish that the sharpness-aware minimization serves as a special instance within their proposed framework. The experimental results demonstrate the usefulness of this framework.

**Strengths:**

This paper considers an important question of neural networks training. It is clearly written and easy to follow. The idea of extending the form of SAM to the DRO framework is natural but very meaningful, since it provides more well-developed tools to understand and improve the generalization of training models.


**Weaknesses:**

1. In the theory, this paper shows that SAM is equivalent to the optimization problem in the OT-MDR framework when using a particular distance metric and picking the center distribution as a Dirac delta distribution. In the experiments, the results show (1) OT-MDR performs better than SAM for both single model and ensemble models cases; (2) The loss landscape found by OT-MDR is more flatten than the one found by SAM. This probably implies the training algorithm of OT-MDR is better than directly using SGD to solve SAM. But the paper doesn’t provide much intuitive understanding or analysis of these observations.

2. Most of the experimental results don’t provide standard deviation, which makes the results less solid.


**Questions:**


Q1. In SAM [1], their experiments show that using 2-Wasserstein distance is better than using \infty-Wasserstein distance. For the current paper, in theorem 4.3, it shows that choosing center distribution as a Dirac delta in OT-MDR with 2-Wasserstein distance is equivalent to SAM (using 2-Wasserstein distance). But based on theorem 4.2, the algorithm solves the minimization problem of OT-MDR with \infty-Wasserstein distance. And the algorithm of OT-MDR still outperforms the SAM (with 2-Wasserstein distance). Does this probably imply the performance of OT-MDR can be further improved when using algorithms that solves the minimization problem of OT-MDR with 2-Wasserstein norm? Here are a few more delicate methods [2,3] to solve the general distributional robustness learning problem.

  [1] Foret, P., Kleiner, A., Mobahi, H., & Neyshabur, B. (2020). Sharpness-aware minimization for efficiently improving generalization.

  [2] Wang, G., & Chizat, L. (2022). An exponentially converging particle method for the mixed nash equilibrium of continuous games.

  [3] Trillos, C. G., & Trillos, N. G. (2023). On adversarial robustness and the use of Wasserstein ascent-descent dynamics to enforce it.


Q2. In figure 1, the difference of accuracy for using different numbers of particles is actually small. Since this figure doesn’t provide the information standard deviation, it might be difficult to conclude that using 2 particles gives the best results. It would be better if the author could provide different random seeds results of this experiment. It will probably provide more insight.

Q3. As mentioned in the weakness, the experimental results in table 1, 3 and 4 don't contain standard deviation. Could the author provide more results on those experiments?


**Limitations:**

In general, I like the idea and the presentation of this paper. The authors could probably further improve this paper through more experiments and add some analysis of the observations from comparing SAM and OT-MDR.

---

> ### Author Rebuttal · Authors · 2023-08-10
>
> We sincerely appreciate your constructive comments. We are dedicated to addressing all the questions listed below to the best of our capabilities.
>
>
> **Q1. In SAM [1], their experiments show that using 2-Wasserstein distance is better. But based on theorem 4.2, the algorithm solves the minimization problem of OT-MDR82 with $\infty$-Wasserstein distance still outperforms the SAM. Does this probably imply the performance of OT-MDR can be further improved when using algorithms that solve the minimization problem of OT-MDR with 2-Wasserstein norm?**
>
>
> We develop our theories for the distance $d(\theta, \tilde{\theta}) = \Vert \theta - \tilde{\theta} \Vert_2^p$. In Theorem 4.2, we propose a tractable solution when $p \rightarrow +\infty$ in which the ball $B_\rho$ and the geometry use the L2 (i.e., Euclidian) distance as in SAM. However, as you mention, the OT-based distributional robustness on a model space and its connection to SAM leave open doors to develop other theoretically-guaranteed and practical methods in the future by considering a more general family of distances between two models or customizing the OT-based distributional robustness itself.
>
> We really appreciate your recommended papers [2,3],  as we believe they are strong tools to develop a more generalized version of this work.
>
>
> **Q2. In figure 1, the difference of accuracy for using different numbers of particles is actually small and doesn’t provide the information standard deviation.**
>
> We would like to update the result of multiple particles as below. Experiments for each number of particles are run three times and report the mean and standard deviation. We agree that the gap between one and two-particle is small but accuracy on two-particle still yields the highest performance consistently.
>
> Table 5. Multiple particle classification accuracies on the CIFAR datasets with WideResnet28x10
> | K particles        |      1       |       2     |         3     |
> | :---                    |    :----:   |    :----:   |     :----:   |
> Accuracy | 83.22 $\pm$ 0.18 | 83.31 $\pm$ 0.16 | 82.95 $\pm$ 0.08
>
> **Q3. The experimental results in Tables 1, 3, and 4 don’t contain standard deviation**
>
> We acknowledge the importance of mean and standard deviation when dealing with distribution robustness problems. Meanwhile, we are able to provide additional results for single models (Table 1 in the main paper), Ensemble models (Table 3 in the main paper), and Gaussian posterior models (Table 4 in the main paper) within the attached PDF. Each experiment is run at least three times with different random seeds. It is worth noting that we managed to re-produce the result of both baselines and our methods for a fair comparison. Due to the time limitation of the rebuttal period, we focus on re-producing experiments with our OT-MDR and will report the remaining baselines in the revised version. Based on the results, our proposed methods consistently demonstrate remarkable performance in most of the experiments. On the other hand, the baseline models, particularly the Gaussian posterior experiment, yield a notably high standard deviation.

---

> ### Comment · Area_Chair_a4y9 · 2023-08-20
> **Reminder from AC**
>
> Dear reviewer,
>
> The author-reviewer discussion period ends in 2 days. Please review the authors' rebuttal and engage with them if you have additional questions or feedback. Your input during the discussion period is valued and helps improve the paper.
>
> Thanks, Area Chair

---

> > ### Comment · Reviewer_NFva · 2023-08-21
> >
> > I would like to thank the authors for their responses. Along with previous recommendations, the experimental results should be updated with standard deviations. Overall I would like to keep the score.

---

> > > ### Author Response · Authors · 2023-08-22
> > >
> > > Thank you again for taking the time to review our paper and give us valuable comments. Due to the rebuttal time constraints, for the standard deviation, we focus on experiments with OT-MDR as reported in the PDF file. We will definitely add full experiments and results to our revised version.

---

### Official Review · Reviewer_P1BN · 2023-07-13

**Soundness:** 3 good
**Presentation:** 4 excellent
**Contribution:** 3 good
**Rating:** 7
**Confidence:** 3

**Summary:**

This paper proposes a novel distributional robustness optimization problem on the space of model parameters (OT-MDR). By formulating the uncertainty set with Wasserstein distance and introducing an entopy regularization, the explicit solution to the primal problem is derived. A practical algorithm is proposed based on sampling techniques from SGLD. The algorithm is extended beyond point estimation to accomodate emsembling and bayesian learning. The dual of OT-MDR turns out as a stochastic relaxation of the objective of SAM, bridging sharpness aware optimization and distributional robustness. Empirical results on Cifar validates the advantage of proposed method over baseline optimizers.

**Strengths:**

1. I might be unfamiliar with literature on sharpness aware optimization, but from my perspective this paper is the first to reveal an intrinsic connection between distributional robustness and local sharpness. Theorem 3.4 is intriguing since it might inspire further research to apply abundant literature on distributional robustness (and adversarial robustness which is highly relative) to solving sharpness aware optimization.
1. The proposed method is a stochatstic extension of SAM but it introduces limited computational burden. On the other hand, the algorithm well extends to bayesian learning because of its stochastic formulation of model parameters.

**Weaknesses:**

This paper proves OT-MDR to be more general than SAM but the advantage of OT-MDR in the case of single model and emsembling is theoretically vague. From Theorem 3.4, it seems that OT-MDR is equivalent to SAM for single model except that OT-MDR softly expands the uncertainty ball around the centering model, by allowing parameters to go beyond the radius $\rho$. The gap is insufficient to account for the empiricial success of OT-MDR for single model. Additionally, is it possible that enlarging the radius $\rho$ for SAM would make its loss landscape approach OT-MDR in Figure 2?

**Questions:**

1. The practical algorithm is based on SGLD which shows that stochastic gradient descent could converge to the posterior distribution of model parameters. However, this paper adopts two-step sampling such that the sampled parameters might deviate much from the distribution in Theorem 4.2. Is this a significant gap between the empirical algorithm and theory?

---

> ### Author Rebuttal · Authors · 2023-08-09
>
> We sincerely appreciate your constructive comments. We are dedicated to addressing all the questions listed below to the best of our capabilities.
>
> **Advantage of OT-MDR in the case of a single model and ensembling**
> In Theorem 4.3, we show a connection of our OT-MDR developed by applying distributional robustness (DR) to model distribution and SAM. This theorem indicates that for a specific distance $d(\theta, \tilde{\theta})$ as in Theorem 4.3, our OT-MDR is reduced exactly to SAM. However, in general, from the modeling perspective, using a more general distance, our OT-MDR can be regarded as a probabilistic relaxation of SAM (cf. Theorem 4.2).
>
> Different from SAM, we cast the problem to sample the particles from a target distribution (i.e., $\gamma^*(\theta, \tilde{\theta})$, which has a closed-form) instead of maximizing the loss using one-step gradient ascent update. Although we observe that the peak of the performance at two or three particles, these multiple-particles versions consume multiplicatively more memory and take multiplicatively longer training time. For a fair comparison, we use the practical version with one single particle to compare to SAM. However,  our particles are sampled in multiple steps using stochastic gradient Langevin dynamics (SGLD). We observe that multiple-step sampling with small Gaussian noises to diversify the particles really helpful to boost the performance. Moreover, randomly splitting a mini-batch into multiple smaller sub-batches, each of which is used for one sampling step, is also beneficial to diversify the particles.
>
> In practice, to balance between performance and computation complexity, we propose using two-step sampling, which was empirically demonstrated to outperform SAM and other baselines as in Tables 1, 2, and 3.
>
> **Enlarging the radius for SAM would make its loss landscape approach OT-MDR in Figure 2**
> In our experiment, for SAM, we use the optimal perturbation radius $\rho$ reported in the original paper.
>
>  Additionally, SAM is sensitive to the perturbation radius $\rho$ as shown in [1]. Please refer to Figure 21 of Appendix E.6 in [1], which shows that the testing accuracies get decreasing when we increase $\rho$ from its optimal values. Moreover, Figure 2 in [1] further indicates that the $m$-sharpness increases if we enlarge $\rho$.
>
>
> **Adopting two-step sampling such that the sampled parameters might deviate much from the distribution in Theorem 4.2: Is this a significant gap between the empirical algorithm and theory?**
>
> The situation here is quite similar to SAM. In the theory of SAM, it suggests minimizing the sharpness
> $
> \max_{\Vert\delta\Vert\leq\rho} \{\mathcal{L}\_{S}(\theta+\delta)-\mathcal{L}\_{S}(\theta)\}
> $
> , which requires multiple gradient ascent steps to find an optimal perturbation. However, in the practical version, it proposes using only one gradient ascent step to find $\delta$. Moreover, Figure 4 in [1] shows the effect of different numbers of projected gradient ascent steps for SAM, indicating that using more projected gradient ascent steps does not necessarily improve the performance.
>
> For our OT-MDR, we adopt two-step sampling using SGLD with two random sub-batches to diversify the particles. We observe that it is sufficient to gain good empirical results, while still balancing the training times. We empirically find that the more-step sampling can slightly boost the performance, but in return more significantly increase the training times. Therefore, we opt the two-step sampling strategy as a practical method.
>
> We conjecture that two-step sampling is good due to an accumulative effect of gradually moving the models $\theta$ or the distribution of models $\theta$ to flatter regions. The two-step sampling particles assist us in moving the models $\theta$ to a flatter region, while the models $\theta$ in a flatter region support the two-step sampling in finding more accurate particles.
>
> [1] Andriushchenko, M. and Flammarion, N. (2022, June). Towards understanding sharpness-aware minimization. In International Conference on Machine Learning (pp. 639-668). PMLR.
>
> [2] Foret, P., Kleiner, A., Mobahi, H., and Neyshabur, B. (2020). Sharpness-aware minimization for efficiently improving generalization. arXiv preprint arXiv:2010.01412.

---

> ### Comment · Area_Chair_a4y9 · 2023-08-20
> **Reminder from AC**
>
> Dear reviewer,
>
> The author-reviewer discussion period ends in 2 days. Please review the authors' rebuttal and engage with them if you have additional questions or feedback. Your input during the discussion period is valued and helps improve the paper.
>
> Thanks, Area Chair

---

### Author Rebuttal · Authors · 2023-08-10

We appreciate the reviewers' valuable comments and would like to report additional experiments in the attached pdf.

---

### Decision · Program_Chairs · 2023-09-21

**Decision:**

Accept (poster)

**Comment:**

The strengths of the paper lie in its theoretical foundation and innovative insights into the connection between distributional robustness and local sharpness. The empirical evidence provided, showing improvements over existing methods, adds further credibility to the work.

While there were initial concerns raised by reviewers about the novelty, presentation clarity, and the absence of standard deviations in the results, the authors addressed these effectively in their rebuttal. They clarified misunderstandings, updated experimental data, and engaged actively in discussions, boosting confidence in their work.

There may still be room for debate on specific aspects of novelty and methodology, but the overall quality and impact of the paper are substantial.

Considering all of these factors, I recommend accepting this paper for publication. Its contributions to the field are significant and warrant its inclusion in the conference.